# Spatial transcriptomics and single-nucleus RNA sequencing reveal a transcriptomic atlas of adult human spinal cord

**Donghang Zhang[1,2†], Yali Chen[1,2†], Yiyong Wei[3†], Hongjun Chen[4†], Yujie Wu[1,2], Lin Wu[1,2], Jin Li[5], Qiyang Ren[6], Changhong Miao[7], Tao Zhu[1], Jin Liu[1,2]\*, Bowen Ke[2]\*, Cheng Zhou[2]\***

[1]Department of Anesthesiology, West China Hospital, Sichuan University, Chengdu, China; [2]Laboratory of Anesthesia and Critical Care Medicine, National-Local Joint Engineering Research Centre of Translational Medicine of Anesthesiology, West China Hospital, Sichuan University, Chengdu, China; [3]Department of Anesthesiology, Longgang District Maternity & Child Healthcare Hospital of Shenzhen City (Longgang Maternity and Child Institute of Shantou University Medical College), Shenhen, China; [4]Department of Intensive Care Unit, Affiliated Hospital of Zunyi Medical University, Zunyi, China; [5]Department of Orthopedic Surgery, Affiliated Hospital of Zunyi Medical University, Zunyi, China; [6]Department of Anesthesiology, Affiliated Hospital of Zunyi Medical University, Zunyi, China; [7]Department of Anesthesiology, Zhongshan Hospital, Fudan University, Shanghai, China

**\*For correspondence:**
scujinliu@gmail.com (JL);
bowenke@scu.edu.cn (BK);
zhouc@163.com (CZ)

[†]These authors contributed equally to this work

**Competing interest:** The authors declare that no competing interests exist.

**Abstract** Despite the recognized importance of the spinal cord in sensory processing, motor behaviors, and neural diseases, the underlying organization of neuronal clusters and their spatial location remain elusive. Recently, several studies have attempted to define the neuronal types and functional heterogeneity in the spinal cord using single-cell or single-nucleus RNA sequencing in animal models or developing humans. However, molecular evidence of cellular heterogeneity in the adult human spinal cord is limited. Here, we classified spinal cord neurons into 21 subclusters and determined their distribution from nine human donors using single-nucleus RNA sequencing and spatial transcriptomics. Moreover, we compared the human findings with previously published single-nucleus data of the adult mouse spinal cord, which revealed an overall similarity in the neuronal composition of the spinal cord between the two species while simultaneously highlighting some degree of heterogeneity. Additionally, we examined the sex differences in the spinal neuronal subclusters. Several genes, such as *SCN10A* and *HCN1*, showed sex differences in motor neurons. Finally, we classified human dorsal root ganglia (DRG) neurons using spatial transcriptomics and explored the putative interactions between DRG and spinal cord neuronal subclusters. In summary, these results illustrate the complexity and diversity of spinal neurons in humans and provide an important resource for future research to explore the molecular mechanisms underlying spinal cord physiology and diseases.

## eLife assessment

Zhang et al. deliver an **important** transcriptomic atlas of the human spinal cord, combining single-cell and spatial transcriptomics to unveil molecular insights. While **convincingly** overcoming Visium limitations using snRNA-seq, the article is criticized for its largely observational approach and lack of quantitative analysis, especially in supporting claims about sex differences in motor neurons and DRG–spinal cord neuronal interactions.

## Introduction

The spinal cord is composed of distinct cell populations, mainly neurons and glial cells (*Abraira et al., 2017*; *Peirs and Seal, 2016*; *Todd, 2017*). The heterogeneity among the various neuronal clusters contributes to the differences in sensory perception, transduction, and processing, as well as the modulation of motor behaviors (*Abraira et al., 2017*; *Bourane et al., 2015*; *Floriddia et al., 2020*; *Hayashi et al., 2018*; *Mishra and Hoon, 2013*; *Todd, 2010*). Recently, single-cell and single-nucleus RNA sequencing (RNA-seq) has provided an unbiased comprehensive strategy to explore gene expression at high resolution and classify cellular clusters in an objective manner (*Aldinger et al., 2021*; *Li et al., 2016*; *Tran et al., 2021*). In particular, single-nucleus analysis has several advantages, such as easy performance in whole tissue and avoidance of experimental artifacts in intact cells that are induced during the tissue dissociation process (*Grindberg et al., 2013*; *Habib et al., 2017*; *Lake et al., 2016*).

Emerging studies have used single-cell or single-nucleus RNA-seq to classify neuronal types in the mouse spinal cord. Using split pool ligation-based transcriptome single-nucleus sequencing (SPLiT-seq), Rosenberg et al. identified 30 neuronal types in the developing spinal cord of mice (*Rosenberg et al., 2018*). Delile et al. performed high-throughput single-cell RNA-seq and revealed the spatial and temporal dynamics of gene expression in the cervical and thoracic spinal cords of developing mice (*Delile et al., 2019*). Using single-cell RNA-seq, Häring et al. identified 15 excitatory and 15 inhibitory subtypes of sensory neurons in the spinal dorsal horn of mice (*Häring et al., 2018*). Sathyamurthy et al. used massively parallel single-nucleus RNA-seq and identified 43 neuronal populations in the lumbar spinal cord of adult mice (*Sathyamurthy et al., 2018*). Russ et al. provided an integrated view of cell types in the mouse spinal cord and their spatial organization and gene expression signatures using single-cell RNA-seq (*Russ et al., 2021*). Recent studies have also provided the single-cell transcriptomic profiles of the developing spinal cord in humans (*Andersen et al., 2023*; *Rayon et al., 2021*; *Zhang et al., 2021*). Although the molecular and cellular organization of the spinal cord neurons is relatively well understood in mice or developing humans, limited research has defined the heterogeneity among neurons and their spatial distribution in the adult human spinal cord at a single-cell resolution, which is important for understanding the molecular basis of spinal cord diseases and physiology because multiple findings from animal experiments cannot be directly replicated in humans (*Kushnarev et al., 2020*; *Yekkirala et al., 2017*).

In this study, we aimed to systematically map the molecular and cellular composition of the adult human spinal cord as well as their spatial location using 10x Genomics single-nucleus RNA-seq and spatial transcriptomics and compare these data with previously published corresponding mouse datasets. This study further examined the sex differences in spinal neuronal clusters to explore the potential mechanism for the differential prevalence of pain between sexes. As a result, we classified human spinal cord neurons into 21 subtypes with distinct transcriptional patterns, molecular markers, and functional annotations. The resulting atlas demonstrates that the overall molecular and cellular organization of the human spinal cord are similar to those reported in mice, although some degree of heterogeneity among transcriptional patterns exists. In addition, we determined that several genes showed sex differences in motor neurons. More importantly, we set up a publicly available website (https://zhangdhscu.shinyapps.io/spinal/) based on our sequencing data, which will be convenient to search genes of interest. This study also used RNAscope in situ hybridization (ISH) and/or immunofluorescence (IF) staining to help validate these conclusions and present evidence for the spatial distribution of several neuronal classes in the human spinal cord. Detailed information on specific gene expression and distribution may serve as an important resource of the cellular and molecular basis for the physiology and etiology of the spinal cord, such as disorders associated with spinal somatic sensation and/or motor behaviors.

## Results

### Identification of spinal cell types by spatial transcriptomics

Using the 10x Genomics Visium Spatial Gene Expression platform, we generated spatial transcriptomes for spinal cells of lumbar enlargements from six adult human donors (three males and three females, 47–59 years old) (*Figure 1A*). Barcoded spots with 55 µm size were printed on the capture areas (6.5 mm × 6.5 mm) in Visium slides. From these spots, we initially identified 17 cellular types

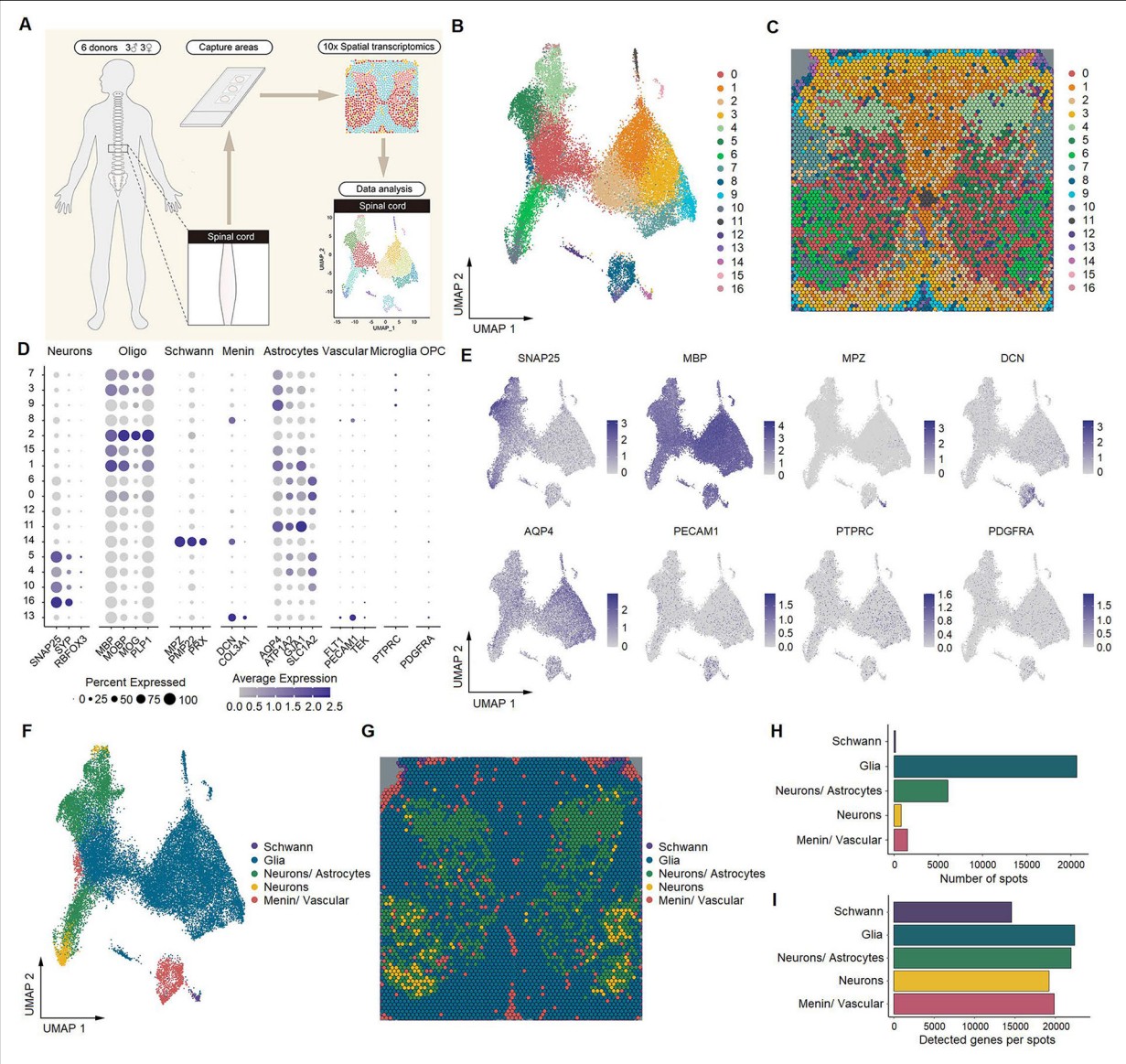

**Figure 1.** Identification of human spinal cell types from spatial transcriptomics. (**A**) Overview of the experimental workflow for spatial transcriptomics. (**B**) UMAP plot showing 17 cell types in the spinal cord. Dots, individual spots; colors, cell types. (**C**) Representative image showing the spatial distribution of these 17 cell types on Visium slides. (**D**) Dot plot showing the expression of representative marker genes across all cell types. The dot size represents the percentage of barcodes within a cluster, and the color scale indicates the average expression across all barcodes within a cluster for each gene shown. (**E**) UMAP plot showing the expression of representative marker genes. (**F**) UMAP plot showing five major cell types in the spinal cord. Glia includes oligodendrocytes, astrocytes, microglia, and OPCs. Dots, individual spots; colors, cell types. (**G**) Representative image showing the spatial distribution of five major cell types in spinal slices. Dots, individual spots; colors, cell types. (**H**) The number of spots in each cluster. (**I**) The number of genes detected per cluster. Oligo, oligodendrocytes; Menin, meningeal cells; OPC, oligodendrocyte precursor cells; UMAP, uniform manifold approximation and projection.

The online version of this article includes the following figure supplement(s) for figure 1:

**Figure supplement 1.** Uniform manifold approximation and projection (UMAP) plot showing the contribution of each donor to spinal cluster formation by spatial transcriptomics.

(*Figure 1B*) with distinct distribution characteristics in the spinal slices (*Figure 1C*). Based on the representative marker genes (*Figure 1D and E*), these 17 cell types were classified into five major clusters, including neurons, Schwann cells, a mixed population of neurons and astrocytes, a mixed population of meningeal and vascular cells, and a mixed population of glial cells (oligodendrocytes, astrocytes, microglia, and oligodendrocyte precursor cells) (*Figure 1F* and G). Because of the lower resolution of

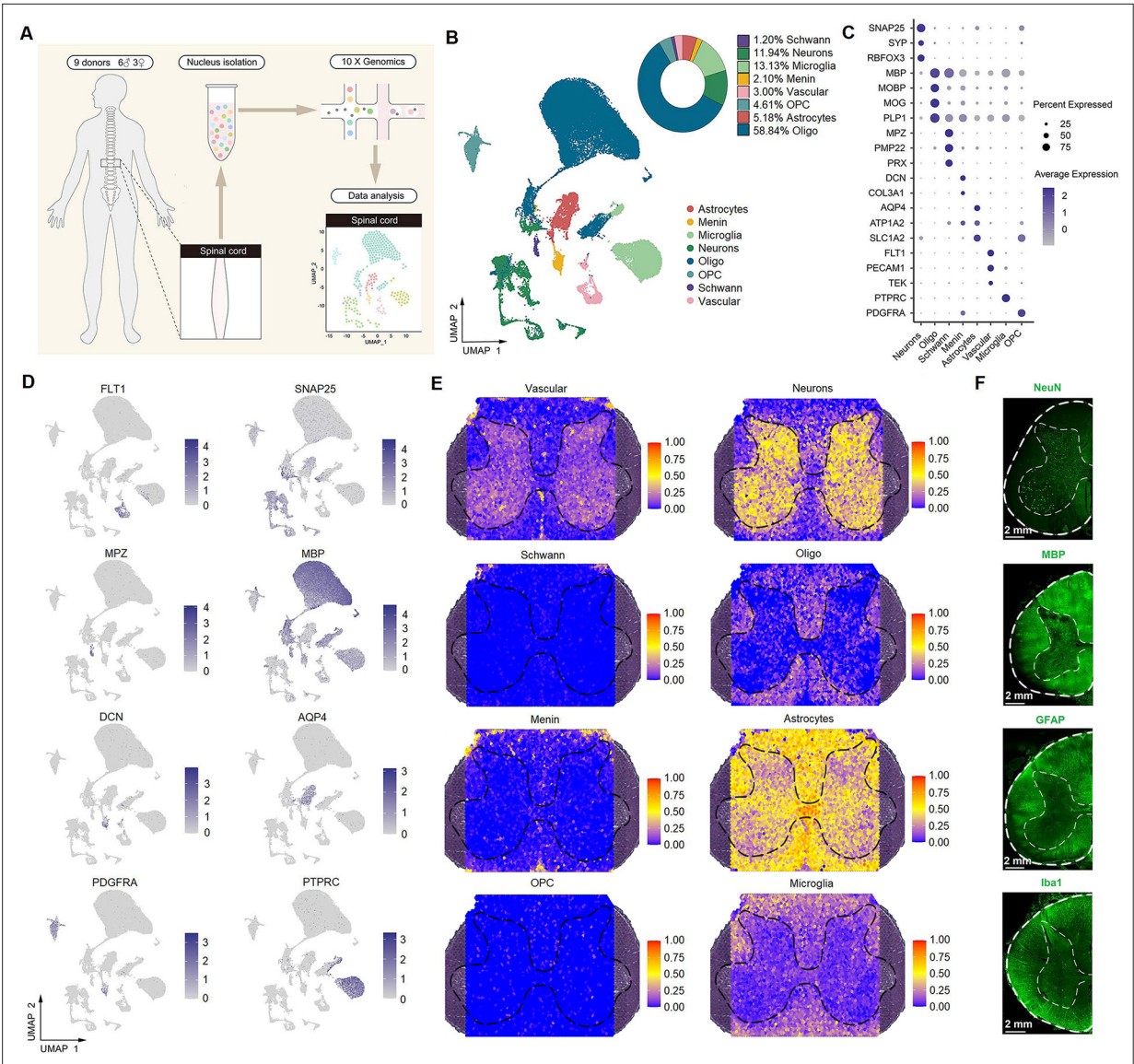

**Figure 2.** Identification of spinal cell types using single-nucleus RNA-seq. (**A**) Overview of the experimental workflow for single-nucleus RNA-seq. (**B**) UMAP plot showing eight major cell types. Dots, individual cells; colors, cell types. (**C**) Dot plot showing the expression of representative marker genes across all eight cell types. The dot size indicates the percentage of cells expressing the gene; the color scale indicates the average normalized expression level in each cluster. (**D**) UMAP plot showing the expression of representative marker genes. (**E**) Representative section showing the spatial distribution of eight clusters in the spinal cord. (**F**) Representative immunofluorescence images of NeuN, MBP, GFAP, and Iba1 in a coronal cryosection of the lumber spinal cord. Oligo, oligodendrocytes; OPC, oligodendrocyte precursor cells; Menin, meningeal cells; UMAP, uniform manifold approximation and projection.

The online version of this article includes the following figure supplement(s) for figure 2:

**Figure supplement 1.** The characteristics of eight cell types in the human spinal cord by single-nucleus RNA-seq.

the spatial transcriptomic approach than that of single-cell or single-nucleus transcriptomes, a large portion of neurons were not well separated from their surrounding astrocytes. The number of spots in these five clusters and genes detected per cluster is shown in *Figure 1H, I*, respectively. We verified that each individual donor contributed spots to each cluster (*Figure 1—figure supplement 1*).

## Identification of spinal cell types using single-nucleus RNA-seq

To obtain single-cell resolution, we further performed single-nucleus RNA-seq on 64,021 nuclei from the lumbar enlargements of the spinal cord from nine adult humans (six males and three females,

35–59 years old), including the same six donors used in spatial transcriptomics (*Figure 2A*). As a result, eight major cell types (*Figure 2B*) were identified with distinct molecular markers (*Figure 2C and D*, *Figure 2—figure supplement 1A and B*): oligodendrocytes (58.8% of total nuclei), microglia (13.1% of total nuclei), neurons (11.9% of total nuclei), astrocytes (5.2% of total nuclei), oligodendrocyte precursor cells (4.6% of total nuclei), vascular cells (3% of total nuclei), meningeal cells (2.1% of total nuclei), and Schwann cells (1.2% of total nuclei). We verified that each individual donor contributed cells to each cluster and that no individual donor was responsible for any specific cluster (*Figure 2—figure supplement 1C*). The number of genes detected per nucleus varied among the major cell types (*Figure 2—figure supplement 1D*). On average, 5245 genes were detected in a single spinal cord neuron, and 2241 genes were detected in a single non-neuron cell. These cell types of single-nucleus RNA-seq were mapped to the spots of spatial transcriptomics to determine their location. The results showed that neurons mainly existed in gray matter, while oligodendrocytes and microglia were mainly present in white matter (*Figure 2E*). Astrocytes were widely present in the spinal cord but were more highly expressed in the white matter than in the gray matter (*Figure 2E*). These findings were validated using IF staining (*Figure 2F*).

## Identification of neuronal subtypes in the spinal cord

To identify and characterize various neuronal subtypes within the spinal cord, 7641 neuronal nuclei were reclassified into 21 subclusters (*Figure 3A*) based on their transcriptional characteristics (*Figure 3—figure supplement 1A and B*). According to the neurotransmitter status, 11 subclusters (54.4%) were defined as excitatory neuronal sets (glutamatergic, with markers of *SLC17A6*), 8 subclusters (34.5%) were inhibitory neurons (GABAergic/glycinergic, with markers of *GAD1*, *GAD2*, and *SLC6A5*), 1 subcluster (9.6%) expressed both excitatory and inhibitory markers, and 1 subcluster (1.5%) was motor neurons (cholinergic, with markers of *CHAT* and *SLC5A7*) (*Figure 3B and D*, *Figure 3—figure supplement 1C and D*).

To propose a relationship between the identified neuronal types and known modality-specific functions, we performed Gene Ontology (GO) term analysis of their top genes. We found that the overall functional assignments for excitatory, inhibitory, and motor clusters were similar in the adult human spinal cord (*Figure 3—figure supplement 2*), such as chemical or glutamatergic synaptic transmission, axon guidance, neuron projection development, glutamate receptor signaling pathway, regulation of NMDA receptor activity, and ion transmembrane transport, suggesting that these neuronal subpopulations cooperated in the regulation of spinal functions.

## Spatial visualization of neuronal subtypes

Next, we identified the distribution pattern of neuronal subtypes by mapping them back to the spatial sections using SpaCET methods as described in a previous study (*Ru et al., 2023*; *Figure 3—figure supplement 3*). To speculate the spatial distribution of each neuronal cluster, the spinal gray matter was separated into three portions, including the superficial dorsal horn (SD, lamina I–II), deep dorsal horn (DD, lamina III–VI), and ventral horn (V, VII–IX), according to previous study (*Todd, 2010*; *Figure 3E*). Excitatory (*Figure 3F*) and inhibitory (*Figure 3G*) neurons were predominantly mapped to the dorsal horn, which was confirmed by IF staining of their markers. Motor neurons (*CHAT* positive) were predominantly located in the ventral horn, with scattered expression in the dorsal horn (*Figure 3H*), which is consistent with the recent published study (*Yadav et al., 2023*). For specific clusters, gene set variation analysis (GSVA) indicated that C3, C4, C6, C7, C8, C10, and C13–19 clusters were predominantly present in the SD; C1, C2, and C12 clusters were in the DD, C5, C9, and C11 clusters were present across the SD and the DD, while C0 and C20 clusters were mainly located in the ventral horn (*Figure 3I*, *Figure 3—figure supplement 4*). We conducted IF staining to validate the expression and distribution of several marker genes in the human spinal cord. For example, C8 was mapped to the superficial dorsal horn, and its representative marker, *PDYN* (prodynorphin, ligands for the kappa-type of opioid receptor), was distributed primarily in laminae I–II (*Figure 3J*), consistent with the previously reported location of *PDYN*-expressing neurons in the rodent spinal cord (*Duan et al., 2014*; *Sapio et al., 2020*; *Serafin et al., 2019*).

Next, we sought to explore the representative marker genes and annotate each cluster. Based on the top three most differentially expressed genes (*Figure 3—figure supplement 1A and B*) and the distribution of classical gene markers in the spinal cord (*Figure 3—figure supplement 5*), we defined

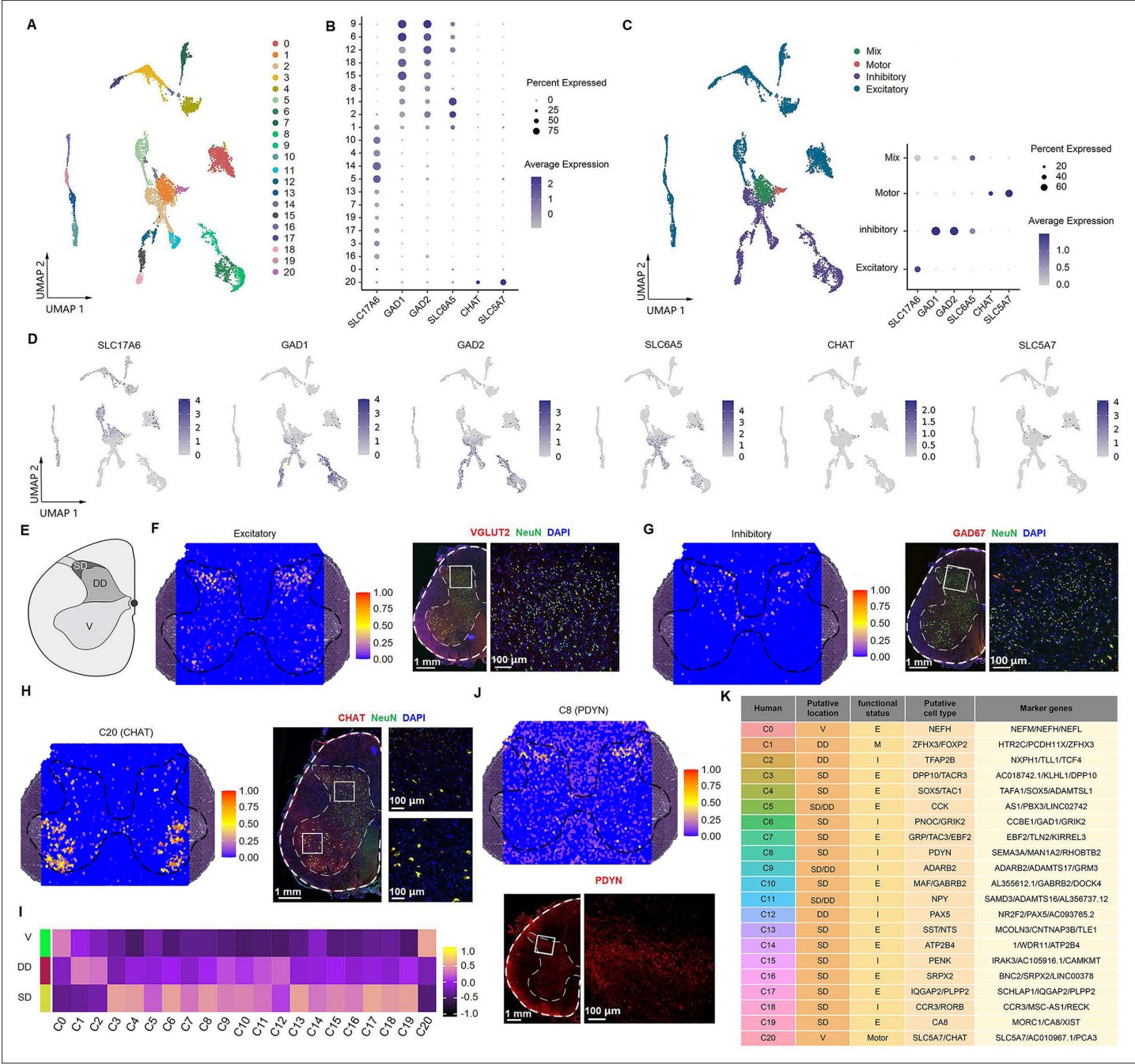

**Figure 3.** Identification of neuronal subtypes in the human spinal cord. (**A**) UMAP plot showing 21 neuronal clusters. Dots, individual cells; colors, neuronal clusters. (**B**) Dot plot showing the expression of selected marker genes across all 21 neuronal clusters. (**C**) UMAP plot of spinal cord neurons based on excitatory, inhibitory, and cholinergic marker genes (left). Dot plot showing the expression of representative marker genes of excitatory, inhibitory, and cholinergic clusters (right). (**D**) UMAP plot showing the expression of representative marker genes of excitatory, inhibitory, and cholinergic clusters. (**E**) Representative cartoons of subregions in coronal slices of the spinal cord. (**F**) Representative section showing the spatial distribution of the excitatory clusters (left) and representative immunofluorescence image of the excitatory (*VGLUT2*) marker in a coronal cryosection of lumber spinal cord (right). (**G**) Representative section showing the spatial distribution of the inhibitory clusters (left) and representative immunofluorescence image of the inhibitory (*GAD67*) marker in a coronal cryosection of lumber spinal cord (right). (**H**) Representative section showing the spatial distribution of motor neurons (C20, left) and representative immunofluorescence image of its marker gene (*CHAT*) in a coronal cryosection of lumber spinal cord (right). (**I**) Gene set variation analysis (GSVA) showing the spatial distribution patterns of neuronal clusters in different subregions of coronal sections from the lumber spinal cord. (**J**) Representative section showing the spatial distribution of C8 (top) and representative immunofluorescence image of its marker gene (*PDYN*) in a coronal cryosection of lumber spinal cord (bottom). (**K**) A summary of the characteristics of neuronal clusters, including their

*Figure 3 continued*

location, functional status, putative neuronal type, and representative marker genes. SD, superficial dorsal horn; DD, deep dorsal horn; V, ventral horn; E, excitatory; I, inhibitory; M, mixed; C, cholinergic; UMAP, uniform manifold approximation and projection.

The online version of this article includes the following figure supplement(s) for figure 3:

**Figure supplement 1.** The gene expression features in 21 neuronal subtypes of human spinal cord.

**Figure supplement 2.** GO term analysis for four functional subpopulations in human spinal cord.

**Figure supplement 3.** The spatial spot showing the distribution pattern of 21 neuronal clusters in the human spinal cord.

**Figure supplement 4.** The spatial distribution patterns of neuronal clusters in different spinal subregions of each donor.

**Figure supplement 5.** UMAP plot showing the expression of representative marker genes in human spinal neuronal clusters.

**Figure supplement 6.** Immunofluorescence results of CCK, SST, and FOXP2.

their putative types and provided the representative marker genes for each neuronal cluster. Finally, we named these clusters by their spatial location and their neurotransmitter status (E, excitatory; I, inhibitory; M, mixed; and C, cholinergic) (*Figure 3K*). We have validated the distribution of several markers using IF staining. For instance, *CCK* (*cholecystokinin, involved in sensory processing under physiological and pain conditions*) (*Liu et al., 2018*; *Peirs et al., 2021*; *Figure 3—figure supplement 6A–D*) and *SST* (*somatostatin, involved in transmitting itch and pain sensation*) (*Fatima et al., 2019*; *Huang et al., 2018*; *Figure 3—figure supplement 6E–H*) were widely present in human dorsal glutamatergic neurons, consistent with the findings in mice (*Todd, 2017*). *FOXP2* (*Forkhead box P2, involved neuronal differentiation and movement disorders*) (*Rousso et al., 2012*) was presented in both human glutamatergic and GABAergic neurons (*Figure 3—figure supplement 6I–L*).

## Human–mouse neuronal-type homology in the spinal cord

To address the similarities and differences in neuronal subtypes between the human and mouse spinal cord, we used Seurat (*Yang et al., 2022*) to anchor the human dataset to the mouse dataset from a previous study (*Sathyamurthy et al., 2018*). Overall, the neuronal data of the human spinal cord overlapped well with those from the mouse in the uniform manifold approximation and projection (UMAP) (*Figure 4A*). To more directly compare the single-nucleus RNA-seq data from human and mouse spinal cord, homologous clusters between humans and mice were identified (*Figure 4B*) using the methods as previously described (*Nguyen et al., 2021*), which showed a highly similar transcriptional profile between species (*Figure 4C*).

We found shared representative genes for several homologous neuronal clusters in humans and mice, such as *ZFHX3* or *Zfhx3 (zinc finger homeobox 3)* for C2, *SOX5* or *Sox5 (SRY-box transcription factor 5)* for C4, and *GRM3* or *Grm3 (glutamate receptor, metabotropic 3)* for C9. We also analyzed the top 20 most differentially expressed genes of excitatory and inhibitory clusters of human and mouse spinal cord (*Figure 4—figure supplement 1*). The transcriptional patterns were similar between human and mouse spinal cords, such as *CHRM3* or *Chrm3 (cholinergic receptor muscarinic 3)*, *NRXN3* or *Nrxn3 (neurexin 3)*, *NXPH1* or *Nhpx1 (neurexophilin 1)*, *DSCAML1* or *Dscaml1 (DS cell adhesion molecule like 1)*, *CACNA2D3* or *Cacna2d3 (calcium voltage-gated channel auxiliary subunit alpha 2 delta 3)*, and *GRIK1* or *Grik1 (glutamate ionotropic receptor kainate type subunit 1)* were predominantly expressed in inhibitory clusters, and *CACNA2D1* or *Cacna2d1 (calcium voltage-gated channel auxiliary subunit alpha 2 delta 1)*, *EBF1* or *Ebf1 (EBF transcription factor 1)*, *EBF2* or *Ebf2 (EBF transcription factor 2)*, *CPNE4* or *Cpne4 (copine 4)*, *CPNE8* or *Cpne8 (copine 8)*, *MAML3* or *Maml3 (mastermind like transcriptional coactivator 3)*, *PDE11A* or *Pde11a (phosphodiesterase 11A)*, *SOX5* or *Sox5 (SRY-box transcription factor 5)*, and *ERBB4* or *Erbb4 (erb-b2 receptor tyrosine kinase 4)* were predominantly expressed in excitatory clusters. *SLC5A7* or *Slc5a7 (solute carrier family 5 member 7)*, *SLIT3* or *Slit3 (slit guidance ligand 3)*, *CREB5* or *Creb5 (cAMP responsive element binding protein 5)*, and *PRUNE2* or *Prune2 (prune homolog 2)* were predominantly expressed in motor neurons. As expected, genes associated with neurotransmitter status were identified (such as *GAD1*, *Gad2*, *Slc6a1*, *Chat*, *SLC5A7*, and *Slc5a7*), but we also observed differential expression between excitatory and inhibitory neurons for a pair of calcium channels (*CACNA2D1* or *Cacna2d1*, and *CACNA2D3* or *Cacna2d3*) in humans and mice. These findings were consistent with a previous study (*Russ et al., 2021*).

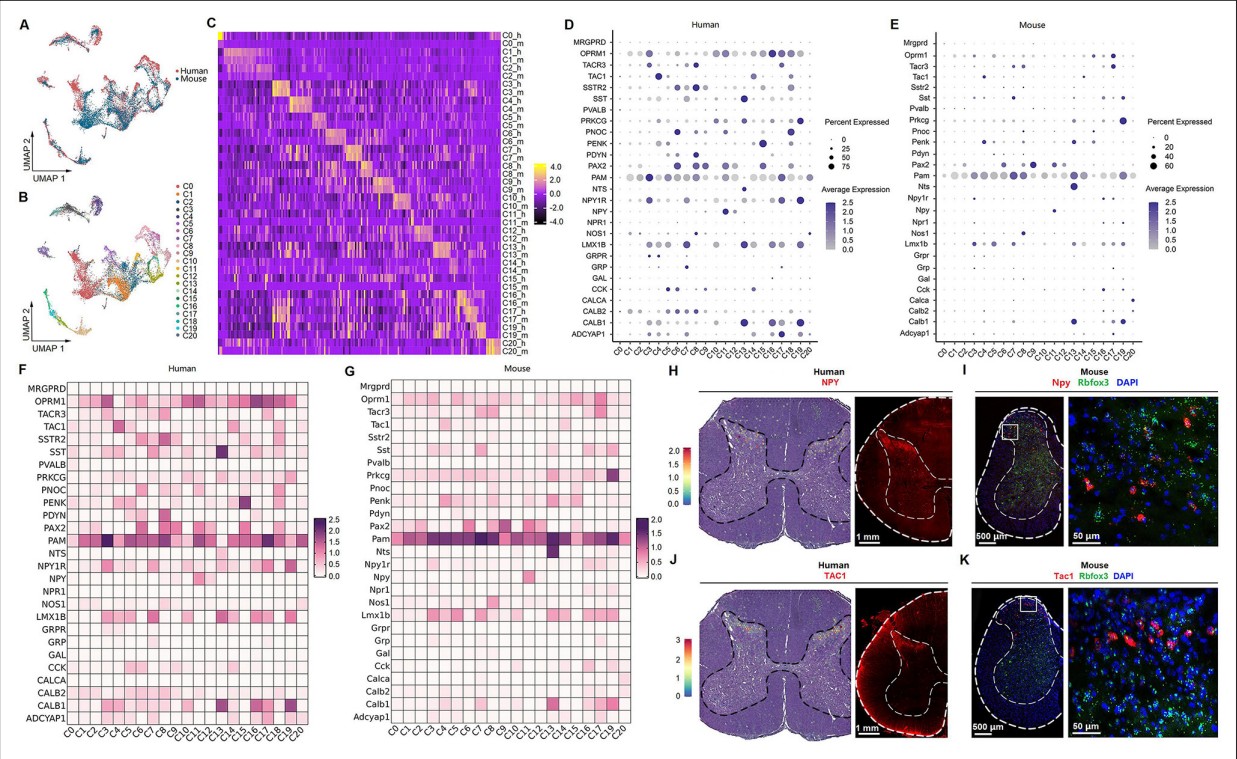

**Figure 4.** Human–mouse cell-type homology. (**A**) UMAP plot showing the coclustering of mouse and human neurons. Dots, individual cells. Colors, species. (**B**) UMAP plot showing the distribution of putative homologous neuronal clusters of humans and mice. Dots, individual cells. Colors, clusters. (**C**) Heatmap of conserved cell-type-specific gene expression (columns) in human and mouse cell types (rows; m, mouse; h, human). Genes in each species are included in the heatmap if they are significantly enriched in a cluster compared to all other clusters (FDR < 0.01, top 50 genes by log$_2$-fold change [FC] per cell type). (**D, E**) Dot plot showing the expression of classical marker genes in the human (**D**) and mouse (**E**) spinal cord. (**F, G**) Heatmap showing the expression of classical marker genes in the human (**F**) and mouse (**G**) spinal cord. (**H**) Representative image showing the distribution of NPY-positive spots in the human spinal cord (left) and representative immunofluorescence image of NPY in a coronal cryosection of the human lumber spinal cord (right). (**I**) Representative RNAscope in situ hybridization images of Npy and Rbfox3 in a coronal cryosection of mouse lumber spinal cord. (**J**) Representative image showing the distribution of TAC1-positive spots in the human spinal cord (left) and representative immunofluorescence image of TAC1 in a coronal cryosection of the human lumber spinal cord (right). (**K**) Representative RNAscope in situ hybridization images of Tac1 and Rbfox3 in a coronal cryosection of mouse lumber spinal cord. UMAP, uniform manifold approximation and projection.

The online version of this article includes the following figure supplement(s) for figure 4:

**Figure supplement 1.** Expression of top 20 most differentially expressed genes in the functional subtypes of human and mouse spinal cord.

**Figure supplement 2.** Expression of classic ion channels and transcription factors in human and mouse spinal neuronal clusters.

**Figure supplement 3.** Expression of selected marker genes in human spinal neuronal clusters.

## Transcriptional profiles of classical markers in spinal cord neurons

To explore the molecular and cellular architecture in the spinal cord neurons and further determine their conservation and divergence between humans and mice, we mapped the expression profiles of classical spinal cord markers in humans and mice, including well-known ion channels, neurotransmitter receptors, neuropeptides, and transcription factors (**Figure 4D–G**, **Figure 4—figure supplement 2**). First, from the atlas, we can make a general comparison of gene expression across different subclusters within humans or mice. The results showed that most genes showed similar expression profiles across different clusters, although several were significantly differentially expressed. For example, the expression of *SST* was significantly higher in C13 than in other clusters of humans (**Figure 4D and F**), and the expression of *NPY (neuropeptide Y, which influences many physiological processes, such as cortical excitability and stress response)* was higher in C11 than in other clusters of humans (**Figure 4D and F**). In the mouse spinal cord, *Nts (neurotensin that may function as a neurotransmitter or a neuromodulator)* and *Nos1 (nitric oxide synthase 1 that functions as a neurotransmitter)* showed selectively higher expression levels in C13 and C8, respectively (**Figure 4E and G**). Second, we also

observed that the overall transcriptional patterns were similar between species. For example, *TAC1* (*tachykinin precursor 1, functions as a neurotransmitter*) and *NPY* were mainly present in the clusters (e.g., C4 and C11, respectively) of superficial dorsal neurons of both human (*Figure 4D and F*) and mouse spinal cord (*Figure 4E and G*), which were confirmed by RNAscope ISH and/or IF staining (*Figure 4H–K*). Nevertheless, some degree of heterogeneity in the expression of specific genes also existed between humans and mice. For example, *Calca* (*calcitonin-related polypeptide alpha, which is involved in migraine*) was selectively expressed in C20 of mice (*Figure 4E and G*), but *CALCA* almost did not exist in human spinal neurons (*Figure 4D and F*). *GRP* (*gastrin releasing peptide*) was predominantly present in C7 of humans (*Figure 4D and F*), but *Grp* showed little expression in all mouse spinal neuronal clusters (*Figure 4E and G*). Of note, several genes, such as *PAM* (*peptidyl-glycine alpha-amidating monooxygenase*), *OPRM1* (*opioid receptor mu 1*), *EBF1*, *GABRG3* (*gamma-aminobutyric acid type A receptor subunit gamma3*), *GRIK2* (*glutamate ionotropic receptor kainate type subunit 2*), *SCN1A* (*sodium voltage-gated channel alpha subunit 1*), *KCND2* (*potassium voltage-gated channel subfamily D member 2*), and *RORA* (*RAR related orphan receptor A*), were highly and widely expressed in the spinal neurons of both humans and mice (*Figure 4D and G*, *Figure 4—figure supplements 2 and 3*), which may provide some guidance for future studies that focus translational efforts on conserved molecular targets across species. However, several genes, such as *CALCA*, *GAL* (*galanin*), and *NPR1* (*natriuretic peptide receptor 1*), showed little expression in the human spinal cord (*Figure 4D and F*, *Figure 4—figure supplement 3J–L*), which should be considered during clinical translation. The expression profiles of each cluster in humans and mice are shown in *Supplementary files 1 and 2*, respectively. In general, this atlas may provide important information for future studies to explore the function of genes of interest in specific subclusters.

## Sex differences in gene expression in spinal neuronal types

The prevalence of chronic pain was higher in females than in males (*Greenspan et al., 2007*), which may result from sex differences in gene expression in the peripheral nervous system (*Avona et al., 2021*; *Mogil, 2020*; *Renthal et al., 2020*; *Sorge et al., 2015*; *Tavares-Ferreira et al., 2022*; *Yu et al., 2020*). However, knowledge of sex differences in gene expression in the adult human spinal cord is limited. Neuronal barcodes from both sexes were represented in all subclusters, suggesting that males and females have the same spinal neuronal subtypes (*Figure 5A*). We then compared gene expression patterns in spinal neuronal subclusters between male and female humans. Genes were considered to be differentially expressed if fold change (FC) ≥ 1.33 and adjusted p<0.05. The comparative results showed similar gene expression profiles between male and female spinal neuronal clusters (*Figure 5B*), except for C20 (motor neurons) (*Figure 5C*). Of note, the most dramatic sex differences in gene expression are known sex-specific genes involved in X inactivation (e.g., *XIST*) or Y chromosome genes (e.g., *UTY*, *USP9Y*), which suggests that the results are reliable.

C20 (motor neurons) had the highest number of differentially expressed genes (DEGs) (*Figure 5C*), suggesting potential molecular differences in mechanisms of motor function between men and women. We were particularly interested in potential sex differences in genes related to pain processing and thus focused on ion channels. A main finding in C20 was the higher expression of genes that encode potassium channels, whereas a lower expression of genes that encode sodium channels was found in males compared to females (*Figure 5D and E*). This finding was validated in IF staining examining *SCN10A* (*sodium voltage-gated channel alpha subunit 10*) expression in *CHAT* neurons from male and female donors (*Figure 5F and G*). In addition, other genes encoding proteins associated with pain sensation were differentially expressed in C20 between male and female donors (*Figure 5E*), such as *HCN1* (*hyperpolarization activated cyclic nucleotide gated potassium channel 1*) (*He et al., 2019*; *Santoro and Shah, 2020*), *FGF13* (*fibroblast growth factor 13*) (*Wang et al., 2021b*), *LRFN5* (*leucine-rich repeat and fibronectin type III domain containing 5*) (*Johnston et al., 2019*), *HTR2C* (*5-hydroxytryptamine receptor 2C*) (*Mickey et al., 2012*), and *KCNQ3* (*potassium voltage-gated channel subfamily Q member 3*) (*Palomés-Borrajo et al., 2021*). GO term analysis of DEGs between males and females showed that upregulated genes in males were associated with pathways implicated in neurotransmitter secretion, synaptic transmission, axon guidance, exocytosis, and neuron project development (*Figure 5H*), while downregulated genes in males were associated with pathways implicated in response to heat, neuron apoptotic process, RNA splicing, and protein catabolic process (*Figure 5I*).

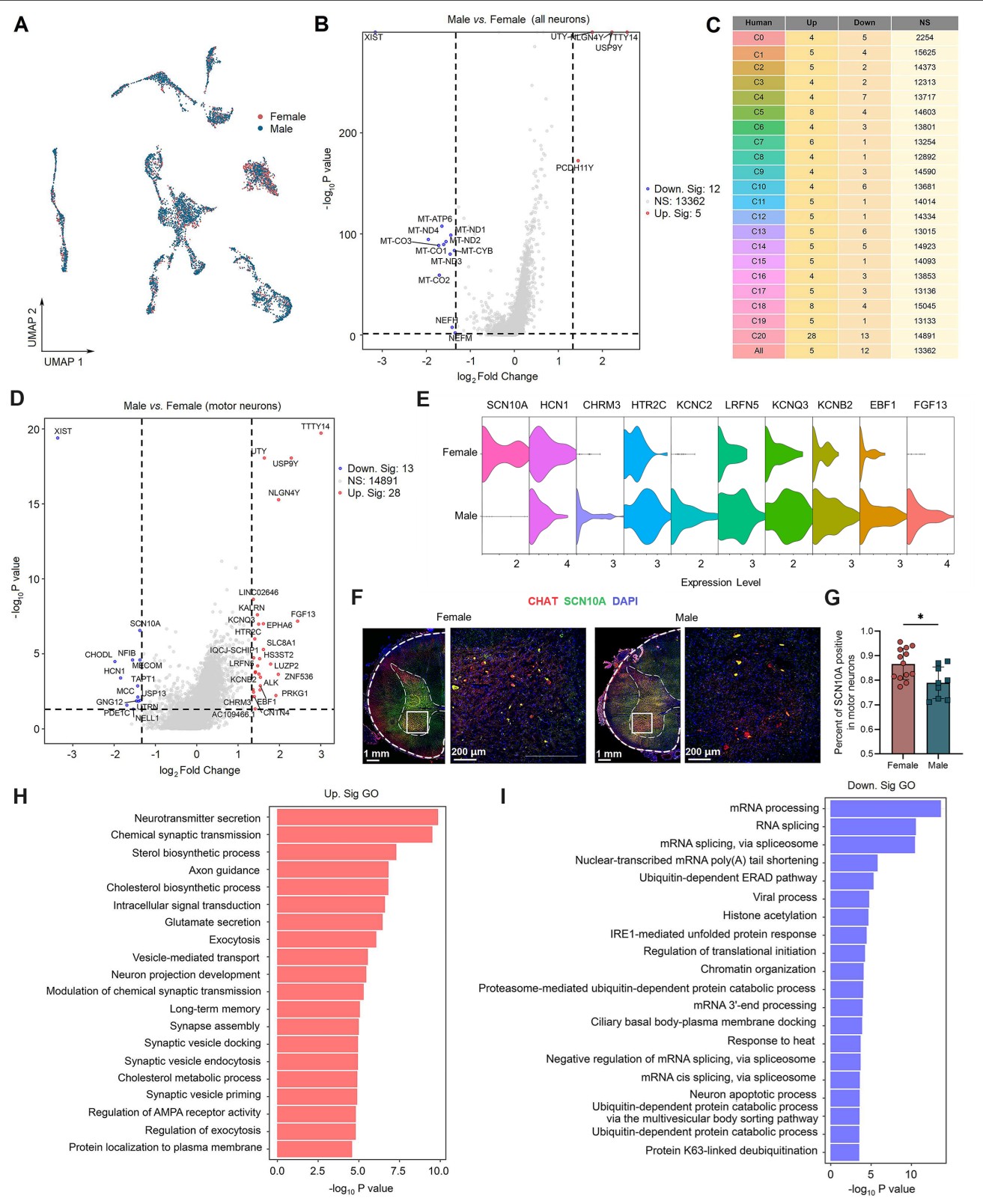

**Figure 5.** Sex differences in gene expression in human spinal neuronal types. (**A**) UMAP plot showing barcodes in all spinal neuronal clusters from males and females. Dots, individual cells. Colors, sexes. (**B**) Volcano plot showing DEGs of all spinal neurons between males and females. (**C**) A summary table showing the number of DEGs for specific neuronal types between males and females. (**D**) Volcano plot showing DEGs in C20 (motor neurons) between males and females. (**E**) Violin plot showing the expression of DEGs in males and females within the motor neurons. (**F**) Representative

*Figure 5 continued on next page*

*Figure 5 continued*

immunofluorescence images of SCN10A with CHAT (a marker of motor neurons) in male and female spinal cord. (**G**) The quantitative analysis of the percentage of CHAT neurons that expressing SCN10A in male and female spinal cord (9 slices from three males and 13 slices from three females, unpaired two-tailed Student's *t*-test, p<0.05). (**H, I**) Summarized GO terms for upregulated DEGs (**H**) or downregulated DEGs (**I**) in male motor clusters compared to females. DEGs were considered if FC ≥ 1.33 and adjusted p<0.05. GO, Gene Ontology; NS, no significance; UMAP, uniform manifold approximation and projection; DEGs, differentially expressed genes.

## Identification of DRG cell types and neuronal subtypes by spatial transcriptomics

We also performed spatial transcriptomics on the L3-5 DRGs from the same six adult donors as in the spinal cord (*Figure 6A*). DRG cells were initially classified into 16 clusters (*Figure 6B*), which were categorized into six major populations based on their distinct transcriptional characteristics (*Figure 6C and D*), including fibroblasts, immune cells, red blood cells (RBCs), Schwann cells, a mixed population of vascular endothelial cells (VECs)/vascular smooth muscle cells (VSMCs), and a mixed population of neurons and satellite glial cells (SGCs) (*Figure 6E*). As expected, neurons and their surrounding SGCs could not be completely separated (*Figure 6E and F*). We verified that each individual donor contributed nuclei to each cluster and that no individual donor was responsible for any specific cluster (*Figure 6—figure supplement 1*).

To determine the heterogeneity within DRG neurons, we further classified them into 13 subclusters: peptidergic neurons (PEP1-6), nonpeptidergic neurons (NP1-2), and neurofilament neurons (NF1-5) (*Figure 7A*, *Figure 7—figure supplement 1A*) based on well-known representative marker genes (*Figure 7—figure supplement 1B*). The number and percentage of barcodes for each cluster is shown in *Figure 7—figure supplement 1C*. We annotated these subclusters according to their top three

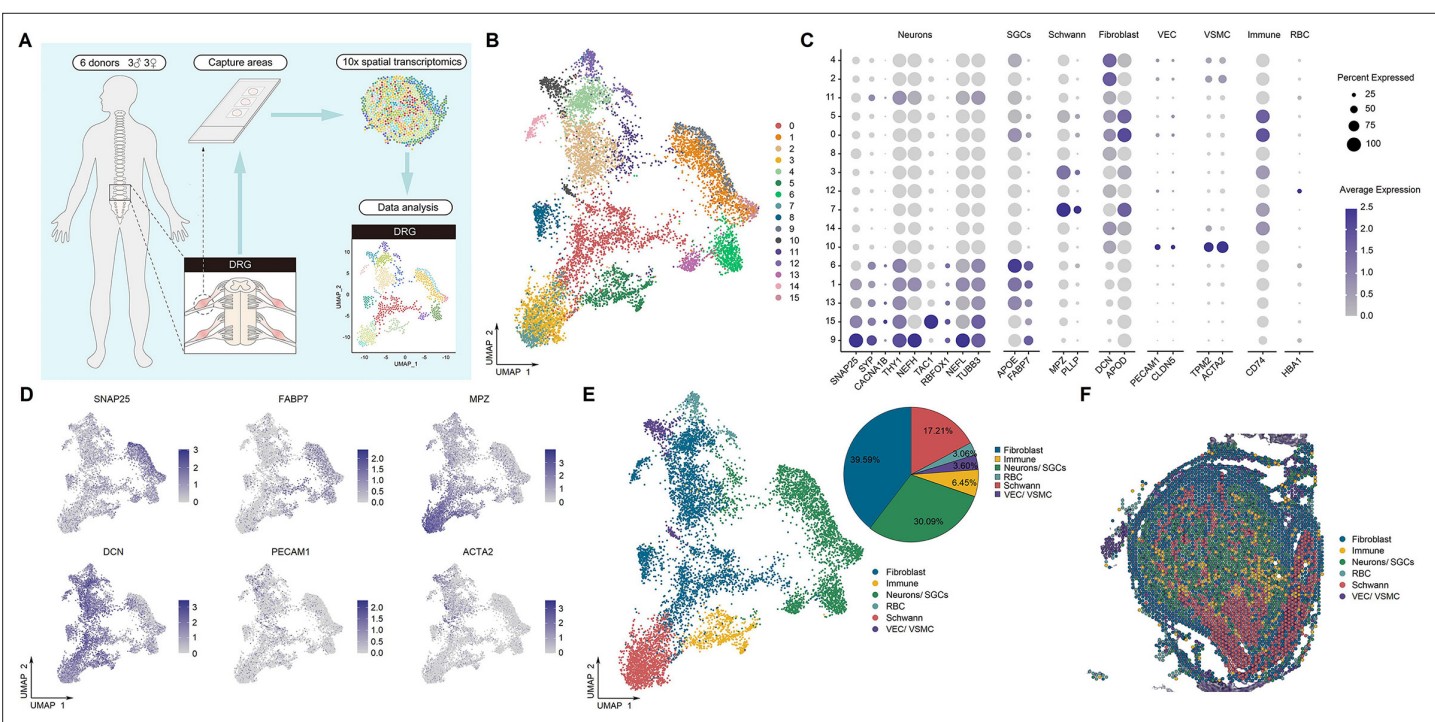

**Figure 6.** Identification of human DRG cell types from spatial transcriptomics. (**A**) Overview of the experimental workflow for spatial transcriptomics in human DRG. (**B**) UMAP plot showing 16 cell types in the spinal cord. Dots, individual spots; colors, cell types. (**C**) Dot plot showing the expression of representative marker genes across all cell types. (**D**) UMAP plot showing the expression of representative marker genes. (**E**) UMAP plot showing six major DRG cell types and their percentages. Dots, individual spots; colors, cell types. (**F**) Representative section showing the spatial distribution of six major cell types in the DRG. Dots, individual spots; colors, cell types. DRG, dorsal root ganglia; RBCs, red blood cells; VECs, vascular endothelial cells; VSMCs, vascular smooth muscle cells; SGCs, satellite glial cells; UMAP, uniform manifold approximation and projection.

The online version of this article includes the following figure supplement(s) for figure 6:

**Figure supplement 1.** UMAP plot showing the contribution of each donor to cluster formation in the DRG by spatial transcriptomics.

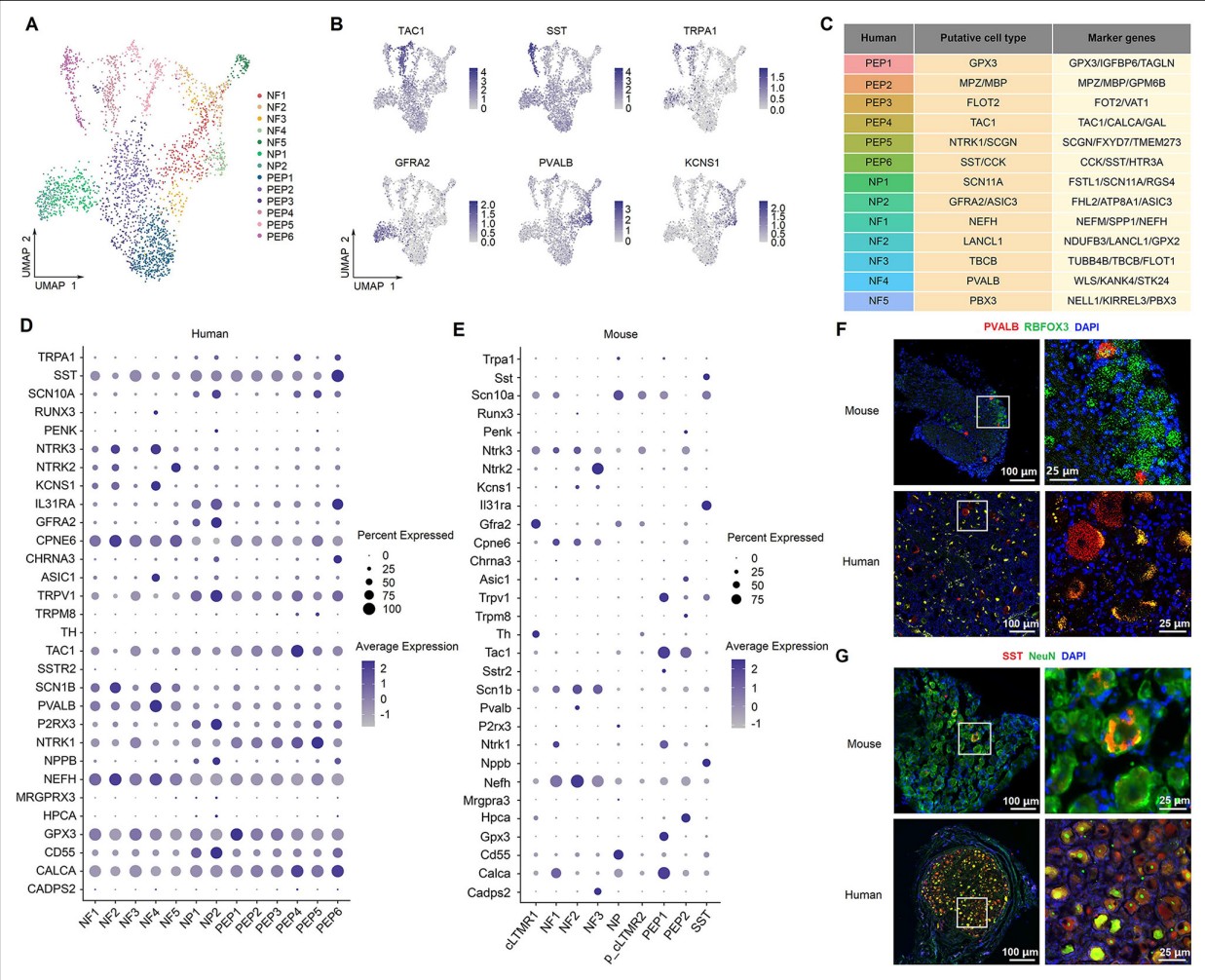

**Figure 7.** Identification of neuronal subtypes in human DRG. (**A**) UMAP plot showing 13 neuronal clusters of human DRG neurons. Dots, individual spots; colors, neuronal clusters. (**B**) UMAP plot showing the expression of representative marker genes. (**C**) Putative DRG neuronal types and their representative marker genes. (**D, E**) Dot plot showing the expression of classical marker genes in human (**D**) and mouse (**E**) DRG neuronal clusters. (**F**) Representative RNAscope in situ hybridization images of PVALB and RBFOX3 in mouse (top) and human (bottom) DRG. (**G**) Representative immunofluorescence images of SST and NeuN in mouse (top) and human (bottom) DRG. DRG, dorsal root ganglia; NF, neurofilament neurons; PEP, peptidergic neurons; NP, nonpeptidergic neurons; UMAP, uniform manifold approximation and projection.

The online version of this article includes the following figure supplement(s) for figure 7:

**Figure supplement 1.** The gene expression features in 13 neuronal subtypes of human DRG.

**Figure supplement 2.** Expression of the top 3 most differentially expressed genes across human DRG neuronal subclusters.

**Figure supplement 3.** Expression of classical marker genes in human and mouse DRG neuronal clusters.

**Figure supplement 4.** Expression of IL4R, IL31RA, and IL13RA1 in human and mouse DRG neuronal clusters.

differentially expressed genes as well as the well-known classical genes of DRG (*Figure 7B*, *Figure 7—figure supplements 1B and 2*). For example, PEP4 was identified as *TAC1/TRPA1* nociceptor clusters, PEP6 was identified as *SST/CCK* clusters (putative silent nociceptors; *Tavares-Ferreira et al., 2022*), NF4 was identified as *PVALB* clusters (putative proprioceptors; *Tavares-Ferreira et al., 2022*), NP2 was identified as *GFRA2* clusters (C-LTMRs; *Tavares-Ferreira et al., 2022*), and NF5 was identified as *NTRK2* clusters (A LTMRs; *Tavares-Ferreira et al., 2022*; *Figure 7C*). We also showed the expression profiles of classical genes in human and mouse (*Renthal et al., 2020*) DRG neuronal subtypes (*Figure 7D and E*, *Figure 7—figure supplement 3*). Consistent with previous studies (*Nguyen et al., 2021*; *Tavares-Ferreira et al., 2022*), our findings indicate a similar transcriptional pattern of DRG neurons between humans and mice. However, our results revealed several genes that showed

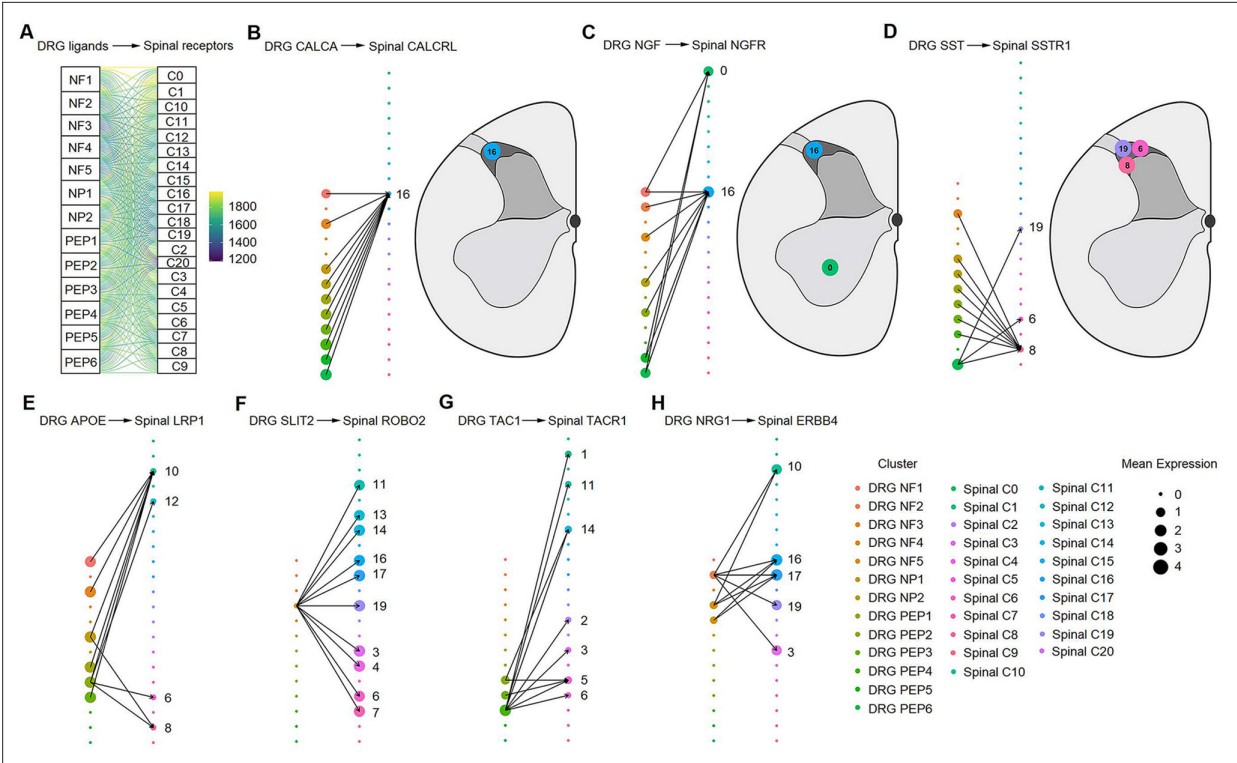

**Figure 8.** Ligand–receptor interactions between the human DRG and spinal cord. (**A**) Putative ligand–receptor interactions of neuronal clusters between the human DRG and spinal cord. The thickness of connecting lines is proportional to the number of total ligand–receptor interactions between the two connecting cell types. (**B**) DRG CALCA-spinal CALCRL interactions (left). The spatial location of the CALCRL-cluster (right). (**C**) DRG NGF-spinal NGFR interactions (left). The spatial location of the NGFR-cluster (right). (**D**) DRG SST-spinal SSTR1 interactions (left). The spatial location of the SSTR1-cluster (right). (**E**) DRG APOE-spinal LRP1 interactions. (**F**) DRG SLIT2-spinal ROBO2 interactions. (**G**) DRG TAC1-spinal TACR1 interactions. (**H**) DRG NRG1-spinal ERBB4 interactions. Dot size denotes relative expression of a gene in each cell type, and colors indicate cell type. Arrows between cell types denote the 10 highest ligand–receptor scores. DRG, dorsal root ganglia; NF, neurofilament neurons; PEP, peptidergic neurons; NP, nonpeptidergic neurons.

different expression between humans and mice. For example, *Sst*-positive neurons represent a distinct subcluster in mouse DRG (*Figure 7E*, *Figure 7—figure supplement 3B*), but *SST* is widely expressed across different DRG subclusters in humans (*Figure 7D*, *Figure 7—figure supplement 3A*). *Th*-positive cells represent a distinct subcluster in mouse DRG (*Figure 7E*, *Figure 7—figure supplement 3B*), but *TH* was almost not present in human spinal neurons (*Figure 7D*, *Figure 7—figure supplement 3A*). *PVALB* is widely expressed in different subclusters of human DRG (*Figure 7D*, *Figure 7—figure supplement 3A*) but shows very low expression in mouse DRG (*Figure 7E*, *Figure 7—figure supplement 3B*). *Gpx3* is selectively expressed in the mouse PEP1 cluster (*Figure 7E*, *Figure 7—figure supplement 3B*), but *GPX3* exhibits high expression in multiple human DRG subclusters (*Figure 7D*, *Figure 7—figure supplement 3A*). We used RNAscope ISH or IF staining to validate these findings by examining *PVALB/Pvalb* (*Figure 7F*) and *SST/Sst* (*Figure 7G*) expression in human and mouse DRGs.

## DRG-spinal cord putative interactions

Next, we explored the putative interactions between classical ligands and receptors within DRG and spinal neuronal subtypes (*Figure 8A*). The ligand *CALCA* from multiple DRG clusters was paired with its receptor *CALCRL* in spinal C16 located in the superficial dorsal horn (*Figure 8B*). *NGF* in DRG neurons was paired with *NGFR* in spinal C0 (ventral horn) and C16 (*Figure 8C*). *SSTR1* in three clusters of the superficial dorsal horn (C6, C8, and C19) was paired with its ligand *SST* in various DRG clusters (*Figure 8D*). *APOE* from several DRG neuronal subclusters was paired with *LRP1* in spinal C6, C8, C10 (superficial dorsal horn), and C12 (deep dorsal horn) (*Figure 8E*). *SLIT2* from NF4 was paired with many spinal subclusters (*Figure 8F*). *TAC1* from PEP2-4 neurons was paired with *TACR1* in several neuronal subclusters of the dorsal spinal cord, including C1, C2, C3, C5, C6, C11, and C14

(*Figure 8G*). *NRG1* from NF clusters was paired with *ERBB4* of several clusters in the superficial dorsal horn, including C3, C10, C16, C17, and C19 (*Figure 8H*). Characterizing ligand–receptor interactions between DRG and spinal cord neurons may be an important future direction for research on sensory transduction.

Taken together, these findings demonstrate the value of our spinal cord atlas as a rich resource for probing the cellular biology underlying complex diseases in the spinal cord.

## Discussion

Increasing evidence from animal studies has demonstrated the diversity and complexity of cellular composition in the spinal cord (*Rosenberg et al., 2018*; *Russ et al., 2021*; *Sathyamurthy et al., 2018*). In the spinal cord, the heterogeneity among various neuronal components makes up complex circuits that process sensory information and regulate motor behaviors (*Abraira et al., 2017*; *Bourane et al., 2015*; *Häring et al., 2018*; *Todd, 2010*). In this study, we profiled the classification of cell types and their spatial information in the adult human spinal cord using 10x Genomics single-nucleus RNA-seq and spatial transcriptomics. As a result, we created an atlas of 21 discrete neuronal subtypes in the human spinal cord. We also defined these neuronal clusters using representative marker genes and identified their spatial distribution. Furthermore, we generated an atlas of the transcriptional profiles of spinal classical genes in the neuronal subtypes. A recent study also provided a cellular taxonomy of the adult human spinal cord using single-nucleus RNA sequencing and spatial transcriptomics (*Yadav et al., 2023*). They classified the cells of adult spinal cord into oligodendrocytes, microglia, astrocytes, neurons, OPC, meningeal cells, endothelial and pericyte cells, lymphocytes, Schwann cells, and ependymal cells (*Yadav et al., 2023*). Consistent with their findings (*Yadav et al., 2023*), we showed oligodendrocytes and microglia were mainly enriched in the white matter, while astrocytes were distributed over the entire spinal cord. They also re-clustered neurons into glutamatergic, GABAergic/glycinergic, and cholinergic groups (motor neurons) (*Yadav et al., 2023*). Using spatial transcriptomics, they showed that dorsal excitatory and inhibitory neurons were mainly localized to the superficial dorsal horn, and the dorsal excitatory and inhibitory neurons were mainly distinguished by neuropeptide genes. For example, the dorsal excitatory neurons were enriched with *TAC1*, *TAC3*, and *GRP*, while the dorsal inhibitory neurons were mainly expressed *PENK*, *PDYN*, and *NPY* (*Yadav et al., 2023*), which were in agreement with our findings. Importantly, our data further identified several genes with sex differences in motor neurons. We also performed spatial transcriptomics in DRG and provided the ligand–receptor interactions between DRG and spinal neuronal types. Therefore, our data not only confirms several findings of a previous study (*Yadav et al., 2023*), and but also will serve as an important supplement in illustrating the complex transcriptomics of adult human spinal cord.

To date, multiple cellular and molecular changes in the spinal cord, including those in the expression of ion channels, neurotransmitter receptors, neuropeptides, and transcription factors, are thought to contribute to pathological pain development after nerve injury (*Yekkirala et al., 2017*). However, it is a great challenge to transfer experimental results from animals directly to humans (*Kushnarev et al., 2020*; *Tibbs et al., 2016*; *Yekkirala et al., 2017*). High variance in cell components and gene expression in individual cell types in the spinal cord among species may partly explain the failure of clinical translation. Although multiple genes associated with pain have been identified in previous studies in rodents, the expression profiles of these genes in neuronal clusters of the human spinal cord remain elusive. In this study, we have provided transcriptional patterns of classical genes associated with pain in different human neuronal clusters. We also compared the differences in the expression of these genes across species. Our results of human-mouse comparisons indicate substantial similarities in the cell components and expression profiles of important genes across species, which is consistent with one recent study (*Yadav et al., 2023*). However, several neuronal clusters showed varied expression profiles in terms of specific genes. For example, *Calca* was selectively expressed in motor neurons of mice, but *CALCA* almost never existed in human spinal neurons. Moreover, we did not identify the mouse homologous cluster for human C18. Interestingly, motor neurons in mouse developing (*Rosenberg et al., 2018*) and adult (*Alkaslasi et al., 2021*) spinal cord, as well as in the human developing spinal cord (*Andersen et al., 2023*), can be divided into two distinct subtypes (alpha and gamma), which were not observed in the adult human spinal cord (*Yadav et al., 2023*). These divergences may partially explain why drug targets in the mouse model cannot be replicated in humans occasionally.

This work reveals the molecular repertoire of each neuronal population, providing a significant extension of our understanding of different cell types in the spinal cord and an important database that allows researchers to probe and analyze the expression profiles of risk genes for human diseases involving the spinal cord. Therefore, this data resource may serve as a powerful tool to advance our understanding of the molecular mechanisms in neuronal populations that mediate spinal cord diseases.

The spinal cord plays a pivotal role in normal sensory processing and pathological conditions, such as chronic pain, providing numerous potential targets for the development of novel analgesics (*Todd, 2010*). However, there is a different prevalence of pain between male and female rodents and humans (*Berkley, 1997*; *Greenspan et al., 2007*; *Mogil, 2020*; *Unruh, 1996*). Previous human genetic studies have attempted to explore the underlying genetic mechanisms and identified several potential targets with sex-dependent expression, such as *OPRM1*, *HTR2A (encoding serotonin 2A receptor)*, *SLC6A4 (encoding sodium-dependent serotonin transporter)*, *P2RX7 (encoding the ATP-gated purinoreceptor P2X7)* (*Mogil, 2020*), and *CALCA* (*Tavares-Ferreira et al., 2022*). However, knowledge about gene expression that exhibits sex differences in the human spinal cord is limited. In our study, we showed that the overall transcriptomic profiles were largely conserved between the male and female spinal cord. Interestingly, for specific neuronal types, we identified several genes associated with pain that were differentially expressed in motor neurons between sexes. For example, *FGF13*, *EBF1*, *KCNB2*, *KCNQ3*, *LRFN5*, *KCNC2*, *HTR2C*, and *CHRM3* are significantly more highly expressed in males than females, while *HCN1* and *SCN10A* are significantly more highly expressed in females than males. Of note, previous evidence showed that the expression of *HCN1* and *SCN10A* exhibit sex differences in several nerve tissues, which might contribute the different prevalence of neurological disorders. For example, Hughes et al. reported that the expression level of HCN1 protein is significant higher (greater than fivefold) in the medial prefrontal cortical of female rats than males (*Hughes et al., 2020*). *SCN10A* in the DRG and spinal cord of rodents is suggested to be associated with the higher prevalence of chronic pain in females compared to males (*O'Brien et al., 2019*; *Paige et al., 2020*). The gene expression differences we observed may suggest potential molecular differences in the mechanisms of fine motor behaviors between men and women.

According to the projection sites of axons, neurons in the spinal cord can be divided into two main types, including projection cells with their axons that project to the brain and interneurons with their axons that remain within the region of the spinal cord (*Todd, 2017*). Both projection neurons and interneurons can be further divided into three functional classes: inhibitory neurons, excitatory neurons, and motor neurons (*Häring et al., 2018*; *Osseward et al., 2021*; *Todd, 2017*). Mouse single-nucleus RNA-seq data revealed that spinal cholinergic clusters (*Chat* and *Slc5a7* markers) belong to excitatory clusters (*Sathyamurthy et al., 2018*). However, both our and previous adult human results (*Yadav et al., 2023*), as well as the developing human data (*Andersen et al., 2023*), showed that cholinergic clusters were independent subpopulations in the human spinal cord. Furthermore, we identified mixed populations that expressed both excitatory and inhibitory markers, which was not consistent with the findings in mice (*Sathyamurthy et al., 2018*). In accordance with a previous study (*Todd, 2017*), our results also showed that some classical marker genes of the spinal cord, including *CCK*, *SST*, *NST*, and *TAC1*, were predominantly present in excitatory neuronal clusters, while others, such as *NPY*, *NOS1*, and *PNOC (prepronociceptin)*, were predominantly present in inhibitory neuronal clusters, which is in line with the recent paper (*Yadav et al., 2023*). A previous study reported that the genes encoding the opioid peptides *Penk (proenkephalin)* and *Pdyn* were expressed by both excitatory and inhibitory interneurons in mice (*Todd, 2017*), whereas our results showed that *PENK* and *PDYN* were predominantly present in human inhibitory neuronal clusters C15 and C8, respectively.

The component heterogeneity of somatosensory neurons in the DRG is essential to relay different peripheral inputs to the central nervous system (*Abraira and Ginty, 2013*; *Gatto et al., 2019*; *Li et al., 2016*; *Todd, 2010*). The heterogeneity in the molecular transcriptomes, neuronal types, and functional properties of DRG neurons in rodents has been well characterized (*Hu et al., 2016*; *Li et al., 2016*; *Renthal et al., 2020*; *Usoskin et al., 2015*; *Wang et al., 2021a*; *Zhang et al., 2022*; *Zhou et al., 2022*). Recently, emerging evidence has revealed the organization of human DRG somatosensory neurons (*Jung et al., 2023*; *Nguyen et al., 2021*; *Tavares-Ferreira et al., 2022*; *Yu et al., 2023*). Using single-nucleus RNA-seq, Nguyen et al. identified 15 subclusters in human somatosensory neurons from lumbar 4–5 DRGs (*Nguyen et al., 2021*). Using spatial transcriptomics,

Tavares-Ferreira et al. defined 12 subpopulations in human DRG sensory neurons, including 5C nociceptors, 1C low-threshold mechanoreceptors (LTMRs), 1 Aβ nociceptor, 2 Aδ, 2 Aβ, and 1 proprioceptor subtype (*Tavares-Ferreira et al., 2022*). Consistent with these results, our study also used spatial transcriptomics to identify several of the same clusters. For example, PEP6, NF4, NF5, and NP2 in our study were matched with putative silent nociceptors, putative proprioceptors, A LTMRs, and C-LTMRs, respectively, in *Tavares-Ferreira et al., 2022*. Jung et al. used single-nucleus RNA-seq to classify human DRG neurons into 11 subclusters, including Proprioceptor & Aβ SA-LTMR, Aβ SA-LTMR Ntrk3, Aδ-LTMR Ntrk2, C-LTMR P2ry1, NP1 Gfra1 Gfra2, NP2 Gfra1, NP3 SST, PEP1 Adcyap1, PEP2 Ntrk1, PEP2 Fam19a1, and Cold Trpm8 (*Jung et al., 2023*). A recent preprint in bioRxiv (*Yu et al., 2023*) used single-soma RNA-seq to identify 16 neuronal types in human DRGs, including hTRPM8, hC. LTMR, hNP1, hNP2, hPEP. SST, hPEP.TRPV1/A1.1, hPEP.TRPV1/A1.2, hPEP. PIEZO, hPEP. KIT, hPEP. CHRNA7, hPEP. NTRK3, hPEP.0, hAδ. LTMR, hAβ. LTMR, hPropr, and hATF3. Some degree of heterogeneity existed in the clusters identified between these studies. For example, *Jung et al., 2023* defined *SST*-positive neurons as NPs, whereas in our study and *Yu et al., 2023*, *SST* clusters belong to PEPs. The H10 cluster in the study of *Nguyen et al., 2021* and the pruritogen receptor cluster in the study of *Tavares-Ferreira et al., 2022* mapped to both NP1 and NP2 in the study of *Jung et al., 2023*. These inconsistent nomenclatures for DRG subtypes across studies influence the comparison and integration of transcriptomic profiles across species, thereby hindering successful clinical translation. Previous studies have identified several genes that are differentially expressed between human and mouse DRG neurons (*Jung et al., 2023*; *Nguyen et al., 2021*; *Tavares-Ferreira et al., 2022*; *Yu et al., 2023*). For example, *SCN4B, IL31RA, IL4R, IL10RA, IL13RA1, TRPV1, SCN8A,* and *CALCA* were more widely expressed across human DRG neuron subtypes than in mice (*Jung et al., 2023*; *Tavares-Ferreira et al., 2022*). As recent evidence indicated that type 2 inflammatory cytokines, such as IL-4, IL-31, and IL-13 could mediate chronic itch by stimulating DRG neurons (*Suehiro et al., 2023*), we explored the expression patterns of these inflammatory cytokines genes in human and mouse DRG. The results showed that *IL4R, IL31RA,* and *IL13RA1* were more widely expressed across human DRG neuron subtypes than in mice (*Figure 7—figure supplement 4*), which is consistent with previous studies (*Jung et al., 2023*; *Tavares-Ferreira et al., 2022*). We additionally found that *PVALB* and *SST* showed broader expression across human DRG neuronal clusters than in mice, suggesting that genes are more selectively expressed in mice than in human DRGs.

It is recognized that dorsal spinal neurons can receive different DRG primary afferents to process different sensory information (*Chen et al., 2020*; *Todd, 2010*; *Todd, 2017*). It is also well known that primary afferent axons terminate in different laminae (*Todd, 2010*; *Todd, 2017*). For example, myelinated low-threshold mechanoreceptive afferents arborize in an area extending from lamina IIi–V, whereas nociceptive and thermoreceptive Aδ and C afferents innervate lamina I and much of lamina II (*Todd, 2010*; *Todd, 2017*). However, we still know little about the interactions between the various neuronal components of the DRG and spinal cord. Wang et al. reported that DRG TRPV1[+] neuron-spinal ErbB4[+] neuron connections are involved in heat sensation and that NRG1-ErbB4 signaling is related to heat hypersensitivity induced by nerve injury (*Wang et al., 2022*). However, which DRG neuronal types secrete NRG1 remains unclear. Previous studies have shown that NRG1 is mainly produced in myelinated neurons and less in TRPV1[+] nociceptors (*Calvo et al., 2010*; *Velanac et al., 2012*; *Willem et al., 2006*). In this study, we explored the potential connections between neuronal clusters in the DRG and spinal cord using ligand–receptor analysis. Interestingly, our results showed that *NRG1* from NF clusters (myelinated DRG neurons) is paired with *ERBB4* of spinal superficial dorsal horn clusters, which is consistent with previous studies (*Calvo et al., 2010*; *Velanac et al., 2012*; *Willem et al., 2006*). Therefore, our findings added evidence that *NRG1* was likely secreted by NF clusters, activated ERBB4[+] neurons in the superficial dorsal horn, and induced heat hypersensitivity. Ligand–receptor analysis also revealed that *NGFR* and *CALCRL* in spinal C16 received broad inputs from various DRG neuronal clusters, suggesting the importance of spinal C16 clusters in DRG–spinal interactions. Therefore, our findings may guide future studies focusing on projections from the DRG to the spinal cord.

This study had several limitations. First, we did not use animal models to explore the function of identified neuronal clusters as well as the functional outcomes of genes with sex differences. Second, the putative projections from DRG to spinal neuronal types were not verified. Last, we only performed spatial transcriptomics in human DRG without snRNA-seq; therefore, the putative projections from

DRG to spinal neuronal types and the comparison results between human DRG spatial transcriptomics data and mouse DRG single-nucleus RNA-seq data should be treated with caution. It is of significance to address these important questions in future studies. It will also be interesting to identify the molecular and cellular heterogeneity in human spinal glial cells, and the similarities and differences across species, although preliminary results were obtained in DRG (*Zhang et al., 2023*).

In summary, this work presents an atlas of adult human spinal neuronal types and their gene expression signatures and will provide an important resource for future research to investigate the molecular mechanism underlying spinal cord physiology and diseases.

## Materials and methods

### Study design

This study aimed to classify human spinal cord neurons and determine their spatial distribution, as well as explore the putative projections between DRG–spinal neuronal clusters using single-nucleus RNA sequencing and spatial transcriptomics. For this purpose, Lumbar enlargements of the spinal cord and L3-5 DRGs were acutely isolated from nine adult brain-dead human donors (six males and three females, 35–59 years old) within 90 min of cross-clamping in the operating room. These donors had not been diagnosed with acute/chronic low back or lower limb pain and mainly died from cerebral hernia or intraventricular hemorrhage. To perform spatial transcriptomics, the spinal cord and DRG tissues were sectioned onto Visium slides, stained, and imaged. Tissue permeabilization and Visium spatial libraries were constructed using a Visium spatial library construction kit (10x Genomics, PN-1000184) according to the manufacturer's instructions. The libraries were finally sequenced using an Illumina NovaSeq6000. We also examined the sex differences in the spinal neuronal subclusters between six male and three female donors. Additionally, we compared the transcriptional profiling obtained in human samples with previously published single-nucleus transcriptomic data of the mouse spinal cord. Finally, several sequencing findings were validated using RNAscope ISH and IF staining. All human procedures were approved by the Ethical Committee of the Affiliated Hospital of Zunyi Medical University on May 19, 2021 (Approval No. KLL-2020–273) and this study was registered with the Chinese Clinical Trial Registry (ChiCTR2100047511) on June 20, 2021. Written informed consent was obtained prior to patient enrollment. C57BL/6J male mice (6~8 weeks old, 20–25 g) were used for animal experiments. All animal procedures were approved by the Animal Ethics Committee of the West China Hospital of Sichuan University (approval ID: 2021420A, Chengdu, Sichuan, China).

### Human sample preparation

Lumbar enlargements of the spinal cord and L3-5 DRGs were acutely isolated from nine adult brain-dead human donors within 90 min of cross-clamping in the operating room. Surgical procedures were performed by the same surgeon from the department of orthopedic surgery. Samples were immediately cleaned from the blood and connective tissue and frozen on dry ice. Spinal cord tissues were cut transversely, parts were embedded in optimal cutting temperature (OCT) compound for spatial transcriptomics, and the remaining tissues were stored at −80°C for single-nucleus RNA-seq, RNAscope ISH, or IF staining. For DRGs, the whole ganglia were embedded in OCT for spatial transcriptomics, and other DRGs were stored at −80°C for RNAscope ISH or IF staining.

### Staining and imaging for spatial transcriptomics

Cryosections were cut at 10 μm thickness and mounted onto the GEX arrays. Sections were placed on a Thermocycler Adaptor with the active surface facing up and were incubated at 37°C for 1 min. Then, sections were fixed with methyl alcohol at –20°C for 30 min, followed by staining with hematoxylin and eosin (H&E) (Eosin, Dako CS701, Hematoxylin Dako S3309, bluing buffer CS702). The brightfield images were taken on a Leica DMI8 whole-slide scanner at 10× resolution.

### Permeabilization and reverse transcription for spatial transcriptomics

Protocols for Visium tissue optimization and spatial gene expression were conducted as described by 10x Genomics (https://10xgenomics.com/) using H&E as the counterstain. Briefly, Visium spatial gene expression was processed using a Visium spatial gene expression slide and reagent kit (10x Genomics, PN-1000184). For each well, a slide cassette was used to create leakproof wells for adding reagents.

Then, 70 µL permeabilization enzyme was added and incubated at 37°C for 12 min to achieve optimal permeabilization. Each well was washed with 100 µL of SSC, and 75 µL of reverse transcription Master Mix was added for cDNA synthesis.

## cDNA library preparation for spatial transcriptomics

At the end of first-strand synthesis, RT Master Mix was removed from the wells. Wells were filled with 75 µL 0.08 M KOH, incubated for 5 min at room temperature, and then washed with 100 µL EB buffer. Each well was supplemented with 75 µL of Second Strand Mix for second-strand synthesis. cDNA amplification was performed on an S1000 Touch Thermal Cycler (Bio-Rad).

Visium spatial libraries were constructed using a Visium spatial library construction kit (10x Genomics, PN-1000184) according to the manufacturer's instructions. The libraries were finally sequenced using an Illumina NovaSeq6000 sequencer with a sequencing depth of at least 100,000 reads per spot with a paired-end 150 bp (PE150) reading strategy (performed by CapitalBio Technology, Beijing).

## Isolation of nuclei for single-nucleus RNA-seq

Nuclei were isolated using a Nucleus Isolation Kit (Cat# 52009-10, SHBIO, China) according to the manufacturer's protocols. Briefly, frozen human spinal cord was thawed on ice, minced, and homogenized in cold lysis buffer containing 1% BSA. Then, the lysate was filtered through a 40 µm cell strainer and centrifuged at $500 \times g$ for 5 min at 4°C. The pellet was resuspended in lysis buffer after removing the supernatant, followed by centrifugation at $3000 \times g$ for 20 min at 4°C. The pellet was then filtered through a 40 µm cell strainer, centrifuged at $500 \times g$ for 5 min at 4°C, and resuspended twice in nuclease-free BSA. Nuclei were stained with trypan blue and counted using a dual-fluorescence cell counter.

## cDNA synthesis for single-nucleus RNA-seq

Nuclei were loaded onto a Chromium single-cell controller (10x Genomics) to generate single-nucleus gel beads in the emulsion (GEM) using a single-cell 3' Library and Gel Bead Kit V3.1 (10x Genomics, 1000075) and Chromium Single Cell B Chip Kit (10x Genomics, 1000074) according to the manufacturer's instructions. Approximately 8040 nuclei were captured from each sample. The captured nucleus was lysed, and the released mRNA was barcoded through reverse transcription in individual GEMs. Reverse transcription was performed to generate cDNA using an S1000TM Touch Thermal Cycler (Bio-Rad) with the following parameters: 53°C for 45 min, 85°C for 5 min, and 4°C until further use. The cDNA was then amplified, and the quality was assessed using an Agilent 4200 (CapitalBio Technology).

## 10x Genomics library preparation and sequencing

The 10× single-nucleus RNA-seq library was established using Single Cell 3' Library and Gel Bead Kit V3.1 according to the manufacturer's instructions. The libraries were sequenced using a NovaSeq6000 sequencing platform (Illumina) with a depth of at least 30,000 reads per nucleus with a paired-end 150 bp (PE150) reading strategy (CapitalBio Technology).

## Single-nucleus transcriptomic data analysis

### Data preprocessing

10x Genomics raw data were processed using the Cell Ranger Single-Cell Software Suite (release 5.0.1) to demultiplex raw base call files into FASTQ files using Cell ranger (v 6.1.2) and then to perform alignment, filtering, barcode counting, and UMI counting using cellranger count. The reads were aligned to the GRCh38 human reference genome using a prebuilt annotation package downloaded from the 10x Genomics website.

### Quality control

Quality control was performed to eliminate low-quality cells, empty droplets, or cell doublets. Low-quality cells or empty droplets usually contain very few genes or exhibit extensive mitochondrial contamination, whereas cell doublets exhibit an aberrantly high gene count. We also detected contamination with low-complexity cells such as RBCs, which are less complex RNA species. Briefly, cells

were filtered out if the gene number was <200 or >10,000,, UMI counts were <500, cell complexity was <0.8, or the mitochondrial gene ratio was >10%.

## Normalization and integration

The Seurat package (v 4.3.0) was used to normalize and scale the single-nucleus RNA-seq data. Data were first normalized by the "Normalize Data" function using the normalization method "Log Normalize'. In detail, the expression of gene A in cell B was determined by the UMI count of gene A divided by the total number of UMIs of cell B, followed by multiplying by 10,000 for normalization, and the log-transformed counts were then computed with base as 2. The top 3000 highly variable genes (HVGs) were detected with the 'Find Variable Features' function using the selection method 'vst'. We then used the 'Scale Data' function to remove the uninteresting sources of variation by regressing out cell–cell variation within gene expression driven by batch, the number of detected UMI, mitochondrial gene expression, and ribosomal gene expression. Finally, the corrected expression matrix was generated for further analysis.

## Dimension reduction, cell clustering, and annotation

We used the 'Run PCA' function in the Seurat package to perform principal component analysis (PCA) on the single-cell expression matrix with genes restricted to HVGs. To integrate cells into a shared space from different batches for unsupervised clustering, the harmony algorithm (v 0.1.0) was used to integrate two batches using the 'Run Harmony' function. We then used the 'Find Clusters' function in the Seurat package to conduct the cell clustering analysis by embedding cells into a graph structure in harmony space. The clustering results were visualized using UMAP. Canonical markers were used to determine the cell-type identity. These include *SNAP25*, *SYP*, *RBFOX3* for neurons; *MBP*, *MOBP*, *MOG*, *PLP1* for oligodendrocytes; *MPZ*, *PMP22*, *PRX* for Schwann cells; *DCN*, *COL3A1* for meningeal cells; *AQP4*, *ATP1A2*, GJA1, *SLC1A2* for astrocytes; *FLT1*, *PECAM1*, *TEK* for vascular cells; *PDGFRA* for oligodendrocyte precursor cells; and *PTPRC* for microglia.

## Differential expression analysis

Differential gene expression analysis was performed using the 'Find Markers' function based on the nonparametric Wilcox rank-sum test for two annotated cell groups. The marker genes were identified using the 'Find All Markers' function in Seurat with settings on genes with at least 0.25 increasing logFC upregulation compared to the remaining cell clusters.

## Enrichment analysis

Gene Ontology (GO) enrichment was conducted using the R package with default settings. GO terms with an adjusted p-value of <0.05, calculated by the hypergeometric test followed by the Benjamini–Hochberg method, were defined as significantly enriched terms. The top 20 enriched terms were selected for visualization.

## Spatial transcriptomics data analysis

We further characterized neuronal clusters by examining their spatial distribution using spatial transcriptomics (ST) data. For ST data, Loupe Browser (v 6.0.0) was used to align the gene expression spots to the image of the tissue slides to ensure a high-quality alignment. Processing of raw reads and calculation of barcode/UMI counts of the ST data were performed using 10x Genomics Space Ranger (v 1.3.1) with GRCh38 as a reference. Ribosomal and mitochondrial genes were removed for further analysis. Filters were applied to keep spots with gene counts ≥30 and UMI counts ≥50. ST data were integrated with Seurat (v 4.3.0). Filtered gene-barcode matrices were normalized with the 'SCTransform' function, and the top 3000 variable genes were identified. Next, we performed PCA, UMAP, and *t*-distributed stochastic neighbor embedding (tSNE) using the first 30 principal components. A shared nearest-neighbor graph was built by the 'FindNeighbors' function in Seurat, and cluster identifications were performed by application of the Louvain algorithm using the 'FindClusters' function in Seurat.

SpaCET (v 1.0.0) was used to conduct cell-type deconvolution of ST data with the above neuronal subcluster datasets as a reference. To identify the spatial location of neuronal subtypes, the spinal gray matter was separated into three regions as described in previous studies, including the superficial

dorsal horn (lamina I–II), deep dorsal horn (lamina III–VI), and ventral horn (VII–IX). GSVA for the selected three subregions was conducted with the R package GSVA (v 1.42.0) using the top 50 positive marker genes as input from neuronal subclusters. A heatmap showing GSVA signature scores per region was visualized with ComplexHeatmap (v 2.14.0).

## Human and mouse neuronal comparison in the spinal cord

Cross-species analysis was performed by comparison of our human findings with mouse data (GSE103892) from a previous study (*Sathyamurthy et al., 2018*). The orthologous genes within the mouse data matrix were converted to human homologs using the R babelgene package (v22.3). Then, label transfer analysis was performed using Seurat (v 4.3.0). Mouse datasets were withheld as the query, while human datasets were used as the reference. Briefly, label transfer analysis consists of two steps. First, the transfer anchors were identified using the 'FindTransferAnchors' function. Second, these anchors were then used to transfer cluster labels to the query dataset with the 'TransferData' function.

## Cellular interaction analysis between DRG and spinal neuronal clusters

Interactions between DRG and spinal neuronal clusters were predicted by ligand receptor pair analysis using SingleCellSignalR (v1.6.0). This package provides its own database of ligand–receptor pairs, and we filtered the ligand–receptor database provided by SingleCellSignalR named LRdb. Data from DRG and spinal cord were first merged using the merge function, and then count data were processed using the LogNormalize function and used as input to predict cell-to-cell interactions. The cell_signaling function was used to predict and calculate the interaction score, and the parameters 'tol' were set to 1, 's.core' was set to 0, and 'int.type' was set to paracrine mode to predict the potential interaction. For each ligand and receptor, cell–cell interactions were plotted for the top 20 highest ligand–receptor scores. Each ligand–receptor pair is scored based on gene expression in every cell type or cluster. Sankey plots were plotted with the R package ggalluvial (v0.12.3) and represent the number of putative cell-to-cell interactions.

## Immunofluorescence staining

Under 2–3% sevoflurane, mice were transcardially perfused with ice-cold Ringer's solution followed by 4% paraformaldehyde. The lumber spinal cord and L3-5 DRGs were extracted and stored in 4% paraformaldehyde overnight, followed by incubation in 30% sucrose for 48 hr. Cryosections from human and mouse spinal cord or DRGs were cut at 12 μm using a freezing microtome (CM1850; Leica, Buffalo Grove, IL) and incubated at 4°C overnight with primary antibodies including NeuN (1:500, mouse, Millipore, Cat# MAB377; 1:500, rabbit, Abcam, Cat# ab104225), MBP (1:300, rabbit, Abcam, Cat# ab218011), GFAP (1:500, guinea pig, Synaptic System, Cat# 173004), Iba1 (1:300, rabbit, Wako, Cat# 019-19741), VGLUT2 (1:500, mouse, Millipore, Cat# MAB5504), GAD67 (1:300, mouse, Santa Cruz Biotechnology, Cat# sc-28376), CHAT (1:300, rabbit, Abcam, Cat# ab181023), PDYN (1:200, rabbit, GeneTex, Cat# GTX113515), NPY (1:200, rabbit, Abcam, Cat# ab22145), TAC1 (1:300, mouse, Abcam, Cat# ab14184), SCN10A (1:200, mouse, Abcam, Cat# AB93616), SST (1:300, rabbit, Abclonal, Cat# A9274), CCK (1:200, rabbit, Thermo Fisher, Cat# PA5-103116), and FOXP2 (1:200, rabbit, Abcam, Cat# ab16046). Then, DRG or spinal sections were incubated with secondary antibodies for 2 hr: goat anti-mouse IgG H&L (Alexa Fluor 488) (1:500, Abcam, Cat# ab150113), goat anti-guinea pig IgG H&L (Alexa Fluor 488) (1:500, Abcam, Cat# ab150185), and Cy3 goat anti-rabbit IgG (1:500, Jackson ImmunoResearch, Cat# 111-165-003). Images were obtained using a Zeiss AxioImager Z.2 (Guangzhou, China) and were processed with ImageJ software (NIH, Bethesda, MD).

## RNAscope ISH

RNAscope ISH was performed following the manufacturer's instructions (Advanced Cell Diagnostics, CA). The following probes were used: Npy (Cat# 313321), Tac1 (Cat# 410351), Pvalb (Cat# 421931), PVALB (Cat# 422181), Rbfox3 (Cat# 313311), and RBFOX3 (Cat# 415591). Images were captured with a NIKON A1R$^+$ two-photon confocal scanning microscope (Shanghai, China) and were processed with ImageJ software (NIH).

## Acknowledgements

The authors thank the organ donors and their families for enduring gift. They also thank Yang Lu from Basebio for providing data analysis assistance.

## Additional information

### Funding

| Funder | Grant reference number | Author |
|---|---|---|
| National Key Research and Development Program of China | 2020YFC2008400 | Cheng Zhou |
| National Natural Science Foundation of China | 81974164 | Cheng Zhou |
| National Natural Science Foundation of China | 82301403 | Donghang Zhang |
| China Postdoctoral Science Foundation | 2021M692276 | Donghang Zhang |
| Natural Science Foundation of Sichuan Province | 2022NSFSC1399 | Donghang Zhang |
| Health Commission of Sichuan Province | 21PJ014 | Donghang Zhang |
| Sichuan Science and Technology Program | 2023ZYD0168 | Cheng Zhou |

The funders had no role in study design, data collection and interpretation, or the decision to submit the work for publication.

### Author contributions

Donghang Zhang, Conceptualization, Data curation, Formal analysis, Funding acquisition, Visualization, Methodology, Writing – original draft, Project administration, Writing – review and editing; Yali Chen, Data curation, Formal analysis, Visualization, Methodology, Project administration; Yiyong Wei, Visualization, Methodology, Project administration; Hongjun Chen, Methodology, Project administration; Yujie Wu, Lin Wu, Jin Li, Qiyang Ren, Changhong Miao, Project administration; Tao Zhu, Supervision, Methodology; Jin Liu, Supervision, Methodology, Writing – original draft; Bowen Ke, Writing – original draft, Writing – review and editing; Cheng Zhou, Conceptualization, Data curation, Formal analysis, Supervision, Funding acquisition, Visualization, Methodology, Writing – original draft, Writing – review and editing

### Author ORCIDs

Donghang Zhang ⓘ http://orcid.org/0000-0002-2754-7204
Yali Chen ⓘ http://orcid.org/0000-0001-5522-0762
Cheng Zhou ⓘ https://orcid.org/0000-0002-5005-2285

### Ethics

Clinical trial registration ChiCTR2100047511.
Human subjects: This study was approved by the Ethical Committee of the Affiliated Hospital of Zunyi Medical University on May 19, 2021 (Approval No.KLL-2020-273) and was registered with the Chinese Clinical Trial Registry (ChiCTR2100047511) on June 20, 2021. Written informed consent was obtained prior to patient enrollment.
All animal procedures were approved by the Animal Ethics Committee of the West China Hospital of Sichuan University (Approval ID: 2021420A, Chengdu, Sichuan, China).

Reviewer #1 (Public Review): https://doi.org/10.7554/eLife.92046.2.sa1

Reviewer #2 (Public Review): https://doi.org/10.7554/eLife.92046.2.sa2
Reviewer #3 (Public Review): https://doi.org/10.7554/eLife.92046.2.sa3
Author Response https://doi.org/10.7554/eLife.92046.2.sa4

## Additional files

### Supplementary files
- MDAR checklist
- Supplementary file 1. The expression profiles of each neuronal cluster of the human spinal cord.
- Supplementary file 2. The expression profiles of each neuronal cluster of the mouse spinal cord.

### Data availability

All data are available in the main text or *Supplementary files 1 and 2*. The accession number for the raw sequencing data and processed data reported in this article is GEO: GSE 243077. The accession numbers for previously published datasets of mouse spinal cord and DRG are GSE103892 and GSE154659, respectively.

The following dataset was generated:

| Author(s) | Year | Dataset title | Dataset URL | Database and Identifier |
|---|---|---|---|---|
| Zhang D | 2024 | Spatial transcriptomics and single-nucleus RNA-sequencing reveal a transcriptomic atlas of human spinal cord | https://www.ncbi.nlm.nih.gov/geo/query/acc.cgi?acc=GSE243077 | NCBI Gene Expression Omnibus, GSE243077 |

The following previously published datasets were used:

| Author(s) | Year | Dataset title | Dataset URL | Database and Identifier |
|---|---|---|---|---|
| Sathyamurthy A, Johnson KR, Li L, Matson KJ, Ryba AR, Bergman TB, Dobrott CI, Kelly MC, Kelley MW, Levine AJ | 2018 | Massively Parallel Single Nucleus Transcriptional Profiling Defines Spinal Cord Cell Types and Their Activity During Behavior | https://www.ncbi.nlm.nih.gov/geo/query/acc.cgi?acc=GSE103892 | NCBI Gene Expression Omnibus, GSE103892 |
| Renthal W, Yang L, Tochitsky I, Woolf C | 2020 | Transcriptional reprogramming of distinct peripheral sensory neuron subtypes after axonal injury | https://www.ncbi.nlm.nih.gov/geo/query/acc.cgi?acc=GSE154659 | NCBI Gene Expression Omnibus, GSE154659 |

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
