## [Editor Report · eLife assessment]

Zhang et al. deliver an **important** transcriptomic atlas of the human spinal cord, combining single-cell and spatial transcriptomics to unveil molecular insights. While **convincingly** overcoming Visium limitations using snRNA-seq, the article is criticized for its largely observational approach and lack of quantitative analysis, especially in supporting claims about sex differences in motor neurons and DRG–spinal cord neuronal interactions.

---

## [Referee Report · Reviewer #1 (Public Review)]

Summary:

Zhang et al. provide valuable data for understanding molecular features of the human spinal cord. The authors made considerable efforts to acknowledge and objectively address the limitations of Visium while attempting to overcome them by utilizing single-nucleus RNA sequencing (snRNA-seq) from the same tissue. By mapping snRNA-seq clusters to Visium data, they offer spatial information, complemented by RNA-ISH and immunofluorescence (IF) validation. They also discuss gender-related differences and the similarities between human and mouse data, aiming to establish a crucial foundation for experimental research. However, I have some comments below.

1. The observation of gender-related differences is interesting. The authors reported that SCN10A, associated with nociceptos, exhibited stronger expression in females. While they intend to validate this finding through IF, the quantitative difference is not clearly observed in the IF data (Figure 5f). It would be essential to provide validation through DAPI-based cell counts, demonstrating the difference in CHAT/SCNA10A co-expression.

2. It is meritorious that in novel features of the transcriptomic study, the authors considered gender-related differences and similarities between humans and mice. Nevertheless, despite the extensive bioinformatics-based analyses performed, the results mostly confirm what has been previously reported (Nguyen et al. 2021; Yadav et al. 2023; Jung et al. 2023).

3. The study did not perform snRNA-seq in the DRG. The limitations of Visium in cell type separation are acknowledged, and the authors are aware that Visium alone has limitations in describing cell expression patterns. The authors need to validate their findings via analyses of public DRG snRNA-seq data (Jung et al. 2023 Ncom; Nguyen et al. 2021eLife) before drawing broad conclusions.

4. Figure 7's comparison between human Visium spot data and Renthal et al.'s mouse snRNA-seq may have limitations as Visium spot data could not provide a transcriptional profile at the single cell resolution. The authors need to clarify this point.

5. Recent findings indicate that type 2 cytokines can directly stimulate sensory neurons. This includes the expression of IL-4RA, IL31RA, and IL13RA in DRG. These findings support the role of JAK kinase inhibitors in mediating chronic itch. Demonstrating the expression of these itch receptors in DRG would be valuable.

6. Given that juxtacrine and paracrine signals operate from 0 to 200 um, spatial information is vital to understanding intercellular communication. The presentation of spatial information using Visium is meaningful, and more comprehensive analyses of potential interaction based on distance should be provided, beyond the top 10 interactions (Figure 8).

7. The gender-related differences are interesting and, if possible, it would be interesting to explore whether age-related differences or degeneration-related factors exist. Using public data could allow the examination of age-related changes.

---

## [Referee Report · Reviewer #2 (Public Review)]

Summary:

In this paper, the authors generated a comprehensive dataset of human spinal cord transcriptome using single-cell RNA sequencing and the Visium spatial transcriptomics platform. They employed Visium data to determine the spatial orientation of each cell type. Using single-cell RNA sequencing data, they identified differentially expressed genes by comparing human and mouse samples, as well as male and female samples.

Strengths:

This study offers a thorough exploration of both cellular and spatial heterogeneity within the human spinal cord. The resulting atlas datasets and analysis findings represent valuable resources for the neuroscience community.

Weaknesses:

The analysis of spatial transcriptomics data was conducted as it is single-cell RNAseq data. However, there are established tools for effectively integrating these two types of data. The incorporation of deconvolution methods could enhance the characterization of each spot's cell type composition.

---

## [Referee Report · Reviewer #3 (Public Review)]

Summary:

Zhang et al sought to use spatial transcriptomics and single-nucleus RNA sequencing to classify human spinal cord neurons. The authors reported 17 clusters on 10x Visium slides (6 donors) and 21 clusters by single-nucleus sequencing (9 donors). The authors tried to compare the results to those reported in mice and claimed similar patterns with some differing genes.

Strengths:

The manuscript provides a valuable database for the molecular and cellular organization of adult human spinal cords in addition to published datasets (Andersen, et al. 2023; Yadav, et al. 2023).

Weaknesses:

The results are largely observatory and lack quantitative analysis. Moreover, the assertions regarding the sex differences in motor neurons and the potential interactions between DRG and spinal cord neuronal subclusters appear preliminary and necessitate more rigorous validation.

---

## [Author Response]

**Reviewer #1 (Public Review):**
Summary:Zhang et al. provide valuable data for understanding molecular features of the human spinal cord. The authors made considerable efforts to acknowledge and objectively address the limitations of Visium while attempting to overcome them by utilizing single-nucleus RNA sequencing (snRNA-seq) from the same tissue. By mapping snRNA-seq clusters to Visium data, they offer spatial information, complemented by RNA-ISH and immunofluorescence (IF) validation. They also discuss gender-related differences and the similarities between human and mouse data, aiming to establish a crucial foundation for experimental research. However, I have some comments below.1. The observation of gender-related differences is interesting. The authors reported that SCN10A, associated with nociceptos, exhibited stronger expression in females. While they intend to validate this finding through IF, the quantitative difference is not clearly observed in the IF data (Figure 5f). It would be essential to provide validation through DAPI-based cell counts, demonstrating the difference in CHAT/SCNA10A co-expression.

Thank you for this important question! We have added panel G in Figure 5, which provided the quantitative analysis of the percentage of CHAT neurons that expressing SCN10A in male and female spinal cord.

1. It is meritorious that in novel features of the transcriptomic study, the authors considered gender-related differences and similarities between humans and mice. Nevertheless, despite the extensive bioinformatics-based analyses performed, the results mostly confirm what has been previously reported (Nguyen et al. 2021; Yadav et al. 2023; Jung et al. 2023).

Thank you! In addition to confirming the findings from previous studies, our results also provided new information regarding the difference between human and mouse. For example, we found that PVALB and SST showed broader expression across human DRG neuronal clusters than in mice, suggesting that genes are more selectively expressed in mice than in human DRGs. Moreover, we identified several genes associated with pain that were differentially expressed in motor neurons between sexes.

1. The study did not perform snRNA-seq in the DRG. The limitations of Visium in cell type separation are acknowledged, and the authors are aware that Visium alone has limitations in describing cell expression patterns. The authors need to validate their findings via analyses of public DRG snRNA-seq data (Jung et al. 2023 Ncom; Nguyen et al. 2021eLife) before drawing broad conclusions.

Thank you for this critical question! It is right that snRNA-seq has a higher resolution in describing cell expression patterns compared to the spatial transcriptomics. We acknowledged the limitation that we only performed spatial transcriptomics in human DRG without snRNA-seq. Nevertheless, our results of spatial transcriptomics in human DRG were similar to previously public snRNA-seq data of human DRG, suggesting a feasibility of using spatial transcriptomics in human DRG.

1. Figure 7's comparison between human Visium spot data and Renthal et al.'s mouse snRNA-seq may have limitations as Visium spot data could not provide a transcriptional profile at the single cell resolution. The authors need to clarify this point.

Thank you! We have clarified this in the limitation section.

1. Recent findings indicate that type 2 cytokines can directly stimulate sensory neurons. This includes the expression of IL-4RA, IL31RA, and IL13RA in DRG. These findings support the role of JAK kinase inhibitors in mediating chronic itch. Demonstrating the expression of these itch receptors in DRG would be valuable.

We have provided the expression patterns of IL-4RA, IL31RA, and IL13RA in human and mouse DRG (Figure 7-figure supplement 4), and cited the relevant paper.

1. Given that juxtacrine and paracrine signals operate from 0 to 200 um, spatial information is vital to understanding intercellular communication. The presentation of spatial information using Visium is meaningful, and more comprehensive analyses of potential interaction based on distance should be provided, beyond the top 10 interactions (Figure 8).

Thank you for this good question! In this study, we focused on the putative projections from DRG to spinal neuronal types, which may be an important future direction for research on sensory transduction. It will be interesting to determine the intercellular communication in the spinal spot using the spatial transcriptomics data in future studies.

1. The gender-related differences are interesting and, if possible, it would be interesting to explore whether age-related differences or degeneration-related factors exist. Using public data could allow the examination of age-related changes.

We agree with the reviewer that it is of great importance to identify the age-related differences using spatial transcriptomics and scRNA-seq data of human spinal cord. However, it is currently difficult to obtain comprehensive results due to the limited human spinal cord datasets regarding different ages.

**Reviewer #2 (Public Review):**
Summary:In this paper, the authors generated a comprehensive dataset of human spinal cord transcriptome using single-cell RNA sequencing and the Visium spatial transcriptomics platform. They employed Visium data to determine the spatial orientation of each cell type. Using single-cell RNA sequencing data, they identified differentially expressed genes by comparing human and mouse samples, as well as male and female samples.Strengths:This study offers a thorough exploration of both cellular and spatial heterogeneity within the human spinal cord. The resulting atlas datasets and analysis findings represent valuable resources for the neuroscience community.Weaknesses:The analysis of spatial transcriptomics data was conducted as it is single-cell RNAseq data. However, there are established tools for effectively integrating these two types of data. The incorporation of deconvolution methods could enhance the characterization of each spot's cell type composition.

Thank you very much for your positive comments and suggestions！Indeed, we have used deconvolution methods to incorporate the spinal snRNA-seq and spatial transcriptomics data.

**Reviewer #3 (Public Review):**
Summary:Zhang et al sought to use spatial transcriptomics and single-nucleus RNA sequencing to classify human spinal cord neurons. The authors reported 17 clusters on 10xVisium slides (6 donors) and 21 clusters by single-nucleus sequencing (9 donors). The authors tried to compare the results to those reported in mice and claimed similar patterns with some differing genes.Strengths:The manuscript provides a valuable database for the molecular and cellular organization of adult human spinal cords in addition to published datasets (Andersen, et al. 2023; Yadav, et al. 2023).Weaknesses:The results are largely observatory and lack quantitative analysis. Moreover, the assertions regarding the sex differences in motor neurons and the potential interactions between DRG and spinal cord neuronal subclusters appear preliminary and necessitate more rigorous validation.

Thank you very much! We have provided the quantitative analysis of the differential expression of SCN10A in male and female spinal cord motor neurons. Our sequencing data revealed putative projections from DRG to spinal neuronal types, which may be an important future direction for research on sensory transduction. We did not use animal models to verify these interactions between DRG and spinal cord neuronal subclusters, which is a major limitation in our study. Nevertheless, our analysis results will provide an important resource for future research to investigate the molecular mechanism underlying spinal cord physiology and diseases.